# Beyond Prediction: Tail-Aware Scheduling for LLM Inference

**Yueying Li** [1]   **Yuanfan Chen** [* 1]   **Jiayang Chen** [* 1]   **Esha Choukse** [2]   **Haoran Qiu** [2]   **G. Edward Suh** [3 4]
**Rodrigo Fonseca** [2]   **Ziv Scully** [5]   **Udit Gupta** [4]

## Abstract

LLM serving exhibits extreme length variability, making size-based scheduling difficult in practice. Recent LLM schedulers approximate SJF/SRPT using predicted decode lengths or rank and primarily report mean-centric metrics (e.g., TTFT/TBT). We show these prediction-driven policies can be fragile under distribution shifts, bursty arrivals, and GPU memory pressure, and still offer limited control over tail latency (P90–P99) that dominates user experience—even with perfect decode-length knowledge. We introduce a distribution-aware, prediction-free scheduling framework that replaces explicit length prediction with soft priority boosting driven by lightweight statistical signals. Our design co-optimizes scheduling with cache-aware preemption to account for memory-coupled decode dynamics that vary across workload mixes. Evaluated on production and open-sourced traces, our method achieves a P99 TTLT up to 35-50% lower than SRPT with perfect length prediction and a TTFT 34-47% lower across various workloads, including reasoning-heavy and chat-heavy tasks, demonstrating a robust alternative for tail-latency optimization in online LLM serving.

## 1. Introduction

Large language models (LLMs) are increasingly deployed in interactive applications such as code generation, conversation, and agentic workloads. In these settings, user experience is governed not only by average performance but also by tail latencies (often defined as 95% or 99% percentile TTFT, TTLT and TBT (Li et al., 2025; Liu et al., 2024; Agrawal et al., 2025; Zhong et al., 2024)), making resource allocation and scheduling an important area to balance throughput and user experience (Duan et al., 2024; Sun et al., 2024; Luo et al., 2025; Zhang et al., 2025b).

Beyond *global* resource allocation like sharding, disaggregation, and routing (Chen et al., 2025; Shi et al., 2025; Patke et al., 2024), modern LLM *local* replica schedulers play an important role in balancing throughput and latency trade-offs by considering the right batch-size, priority, and eviction order (Agrawal et al., 2024a;b). Recently, more advanced scheduling systems in LLM often adopt *prediction-based scheduling*, motivated by classical results showing that Shortest Job First (SJF) or Shortest Remaining Processing Time (SRPT) minimize *mean* response time when job sizes are known (Qiu et al., 2024; Fu et al., 2024; Shahout et al., 2024; Srivatsa et al., 2024). They predict output length or the number of remaining tokens and approximate SJF/SRPT in practice. While effective at reducing average latency under accurate prediction, these methods are largely optimized for mean-centric metrics, and policies that optimize for the mean can exhibit poor tail latency behaviors (Nair et al., 2010; Nuyens et al., 2008; Wierman and Zwart, 2012).

This challenge is amplified by the rise of *reasoning-augmented LLMs*, where generation may involve multi-step reasoning, self-reflection, tool invocation, or adaptive termination. As a result, decode lengths are highly variable. Two requests with identical prefill sizes may diverge by orders of magnitude in completion time, and the job-size distribution can shift rapidly across workloads and over time. This inherent unpredictability makes it difficult to design schedulers that rely on job size estimation or request ranking.

These observations raise a key question: *do schedulers need predictions of job lengths or rankings to optimize tail latency?* Rather than attempting to predict decoding length or ranking, we argue that scheduling should be guided by running statistics and optimized for the tail adaptively.

Our approach is inspired by recent advances in *tail-optimal scheduling theory*, which studies how to reduce asymptotic tail waiting times (e.g., P95/P99). A key takeaway is that, instead of *hard* size-based ranking (e.g., SJF/SRPT), one

---

[*]Equal contribution (co-second authors).   [1]Cornell University, Computer Science Department, NY, USA [2]Microsoft Azure System Research, WA, USA [3]NVIDIA Corporation, CA, USA [4]Cornell University, Electrical and Computer Engineering Department, NY, USA [5]Cornell University, Operations Research and Information Engineering Department, NY, USA. Correspondence to: Yueying Li <yl3469@cornell.edu>.

*Proceedings of the 43$^{rd}$ International Conference on Machine Learning*, Seoul, South Korea. PMLR 306, 2026. Copyright 2026 by the author(s).

should use a policy that, roughly speaking, takes first-come first-served as a baseline and then applies *soft, continuous* priority shaping to improve tail performance. Specifically, each request receives a smoothly varying score, called its "boost", that is computed from lightweight signals, and requests are served in order of increasing arrival time minus boost. Such "boosting" rules can provably suppress extreme delays under partial observability because they gently favor requests likely to dominate the tail without requiring accurate length prediction unless size information is available (Yu and Scully, 2024; Harlev et al., 2025; Charlet and Van Houdt, 2025; 2026; Brooker, 2022; Grosof et al., 2021).

However, applying such policies directly to LLM serving is challenging. Unlike classical queues, LLM inference is *stateful and memory-coupled*: ongoing requests in decode phases accumulate large key–value (KV) caches, making preemption and eviction expensive (Zhang et al., 2025a; Kwon et al., 2023). Naïvely prioritizing requests without accounting for limited KV cache capacity and swapping cost can reduce queueing delay while *increasing TTLT* due to recomputation and cache thrashing.

To address this gap, we propose a *co-design of scheduling and eviction* for online LLM serving. Our approach uses lightweight, observable signals (e.g., decoded token length, past request latency distribution) to drive a *distribution-aware boosting policy*, while jointly managing KV-cache eviction so that prioritization decisions translate into real TTLT improvements. By coupling boost-based scheduling with eviction-aware execution control, we enhance vanilla boosting policies to remain effective under memory constraints and multi-phase execution. Importantly, this design optimizes *TTLT holistically*, balancing short requests and long requests, recomputation costs, and request localities rather than focusing on a single phase or metric.

Empirically, we show that this co-designed approach achieves consistent improvements in *tail TTLT* and *throughput (tokens/s)* across heterogeneous workloads, even when predictor quality degrades (to trade off more scability and less overhead) or distribution shifts occur. These results suggest that *distribution-aware, eviction-conscious scheduling* provides a robust alternative to prediction-heavy SRPT approximations, better aligned with the realities of reasoning-augmented LLM serving.

## 2. Background and Motivation

### 2.1. LLM Generation Length Prediction

**Related Work: Size-Based Scheduling for LLM Serving.** Motivated by classical results that SJF/SRPT minimize mean response time when job sizes are known, recent LLM serving systems adopt size-based prioritization using predicted output length. Fu et al. propose a learning-to-

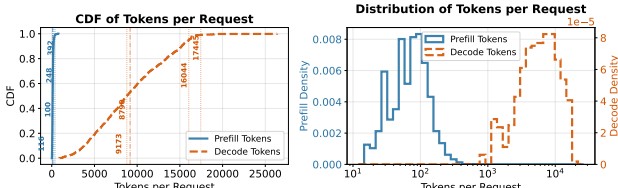

*(a)* Prefill and decode token distribution for s1k workload (Muennighoff et al., 2025)

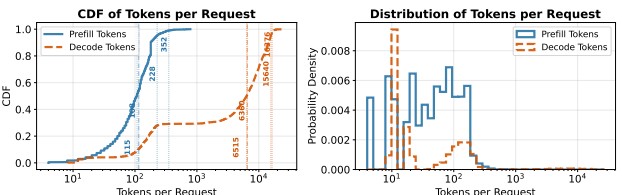

*(b)* Mixture of reasoning and non-reasoning workloads of 10k samples in total used in evaluation (Microsoft Azure, 2024)

*Figure 1.* Distribution of different requests, the four vertical lines show mean, median, P95, and P99, respectively. Prefills are generally light-tailed; for decode length distribution, (a) is light-tailed, while (b) is more heavy-tailed.

rank (LTR) scheduler that approximates SJF by predicting generation lengths, demonstrating improvements in mean and P90 TTFT and time-between-token latencies (Fu et al., 2024). Related approaches such as TRAIL (embedding-based schedulers) similarly rely on learned predictors to estimate remaining execution tokens and guide preemptive scheduling decisions, with a threshold to help reduce preemption at a later phase (Shahout et al., 2024). Heuristic methods in SGLang and vLLM, such as Shortest Prefix First (SPF), avoid explicit output prediction by using the known prefill length as a proxy for the total job size but fail to maintain high throughput or low TTLT.

**Challenges in LLM Output Length Predictability.** In Figure 2, we benchmarked the token-generation variance of the Qwen-2-7B and DeepSeek-R1-Distill Llama models across conversational (WildChat), code (BigCodeBench), and reasoning (S1K) datasets, finding that output variability is consistently high and task-dependent (excluding some system-context-limited output). Across the 2 models × 3 datasets × 3 randomly sampled prompts experimental matrix (Figure 2), we observed high Coefficient of Variation (CV) across reasoning or non-reasoning tasks. For conversational tasks (WildChat), both Qwen ($CV \in [0.10, 0.35]$) and DeepSeek ($CV \in [0.22, 0.47]$) exhibited consistently high natural variance. Code generation (BigCodeBench) showed moderate stability for Qwen ($CV \in [0.10, 0.20]$) but significantly higher instability for DeepSeek ($CV \in [0.07, 0.46]$). In contrast, reasoning tasks (S1K) triggered extreme behaviors: while some prompts led to variable thought processes, many were constrained by loopy behavior up to system constraints, deterministic refusal, resulting in lower variance (Qwen: $CV \in [0.00, 0.19]$, DeepSeek: $CV \in [0.00, 0.12]$).

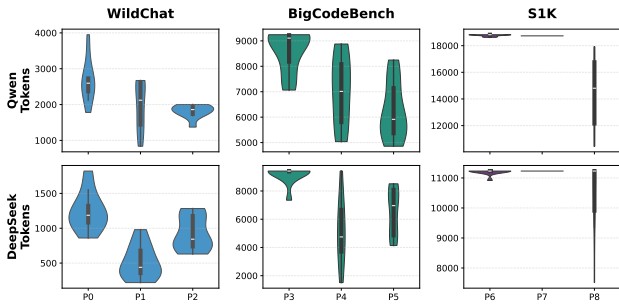

*Figure 2.* Output token across two models of the same randomly chosen set of prompts from three datasets, each prompt running 20 times on SGLang with sampling temperature 0.6 (Zhao et al., 2024; Zhuo et al., 2024; Muennighoff et al., 2025).

The high variance is consistent across all sampled requests and unconditional on the chosen positive temperature.

> **Takeaway**
>
> The significant output length variance observed across all tasks—driven by both natural stochasticity in the sampling process and numerical nondeterminism–makes predicting model output length fundamentally difficult, even when using the precise same prompt and model configuration (Yuan et al., 2025; He and Lab, 2025).

**Metric Mismatch: Mean vs. Tail-Optimality in LLM Serving.** LLM serving is typically *SLO-constrained*: the sustainable load of a serving stack is set by whether it can keep *tail* latencies within budget under burstiness and heterogeneous request lengths (Kaffes et al., 2019; Yu et al., 2022; Zhong et al., 2024; Zhang et al., 2025a; Li et al., 2024; Patel et al., 2023; Goel et al., 2025). Accordingly, **tail completion time**—*Time-To-Last-Token* (TTLT) at P95/P99—directly captures user-perceived responsiveness for end-to-end tasks and often becomes the binding constraint in practice. In contrast, focusing on *mean* TTFT/TBT can be misleading. First, token-level means do not compose: small per-token slowdowns can accumulate over long generations, yielding acceptable mean TTFT/TBT but poor P95/P99 TTLT. Second, mean metrics hide *distributional pathologies*: policies that favor short or partially served requests can cause starvation or head-of-line blocking, inflating tail TTLT despite benign mean TTFT.

Despite this, tail metrics (e.g., P95/P99 TTFT/TTLT) are often under-reported in evaluations of size- or prediction-driven schedulers (Fu et al., 2024; Shahout et al., 2024). More fundamentally, mean-optimality does not imply tail-robustness: when response lengths are capped and some traces become closer to light-tailed (e.g., Fig. 8a), size-based prioritization can be *arbitrarily bad*. In fact, PS and SRPT can achieve the worst possible sojourn-time decay rate (same as PLCFS) under mild conditions (Nuyens et al.,

2008; Nair et al., 2010; Wierman and Zwart, 2012).

## 2.2. Tail-Optimal Scheduling Policy

**Related Work: Tail-Optimal Scheduling in Queuing Theory.** A rich body of queueing-theoretic work studies scheduling policies that optimize tail behavior of response times. For systems with heavy-tailed job sizes, highly preemptive policies with relatively simple priority rules, such as least-attained-service, are known to achieve favorable tail properties without requiring job size information (Nuyens et al., 2008; Scully et al., 2020; Scully and van Kreveld, 2025). The case of light-tailed job sizes, however, is more subtle: recent work shows that optimizing tail latency in light-tailed queues requires a "softer" priority rule. For instance, Yu and Scully (2024) propose a policy called $\gamma$-*Boost* that serves jobs in order of increasing "boosted arrival time", which is a job's arrival time minus a "boost" quantity that is larger for shorter jobs. The effect is that short jobs are given some priority, but not enough to overage longer jobs that have been waiting much longer. Harlev et al. (2025) show that the same Boost design framework can be used for jobs with unknown sizes, too.

Our contribution is to *implement a practical version of $\gamma$-Boost for LLM inference*. However, applying tail-optimal policies for LLM serving faces additional challenges.

**Challenge #1: Need for Distribution-aware LLM Schedulers.** We studied the distribution of traces in different LLM inference datasets (Fig. 1). To analyze scheduling taxonomies in a vLLM-style *chunked-prefill, continuous-batching, decode-prioritized scheduler*, we use various reordering policies both in simulation and in the system. We find that no single scheduler dominates, as the optimal strategy is highly sensitive to the **job size distribution** and **arrival burstiness**. The high variance in job length distribution makes the scheduling problem significantly harder: policies must strictly trade off between mitigating Head-of-Line (HoL) blocking (requires preemption of longer jobs) and preventing starvation (requires protecting long jobs), to optimize for SLO attainment ratio (defined later) or tail latencies while protecting the throughput. Fig 3 visualizes these trade-offs with colors denoting batch mixture.

In Fig. 3 **Left** scenario, we demonstrate HoL blocking driven by job sizes. A long request $A$ (Decode=8, Prefill=2) arrives at $t = 0$, delaying staggered short requests $B, C, D, E$ (Decode=1-2, Prefill=2). FCFS performs poorly here with mean latency inflation, while SRPT/LAS preempt $A$. Conversely, the **Right** scenario (Chunk Size=3) shows how a different job size distribution can inversely harm tail latency. A burst of short jobs $B, C, D$ (Decode=2, Prefill=3) arrives simultaneously or shortly after $A$. The strict preference for short jobs by SRPT/LAS (least-attained-service) causes starvation of $A$, inflating tail (max) latency, whereas FCFS

*Table 1.* Comparison of scheduling approaches for LLM inference. Size-based schedulers (e.g., SJF/SRPT) rely on prediction and primarily optimize mean latency, while tail-aware approaches better capture tail end-to-end latency (TTLT).

| Policy / Work | Heavy-tailed | Light-tailed | No Size Prediction | Preemption Overhead | Tail-Aware by Design | TTLT Evaluated |
|---|---|---|---|---|---|---|
| Shortest Prefix First - SPF (feature in vLLM) | △ | △ | × | × | × | × |
| Rank-prediction-based SJF (LTR) (Fu et al., 2024) | ✓ | △ | × | △ | × | × |
| Prediction-based SRPT (TRAIL) (Shahout et al., 2024) | ✓ | △ | × | △ | × | ✓ |
| FCFS (vLLM) | × | ✓ | ✓ | ✓ | × | △ |
| Skip-Join MLFQ / LAS (Wu et al., 2023) | △ | △ | ✓ | △ | ✓ | △ |
| Boost / $\gamma$-Boost | × | ✓ | ✓ | ✓ | × | △ |
| **Our Work** | △ | ✓ | ✓ | ✓ | ✓ | ✓ |

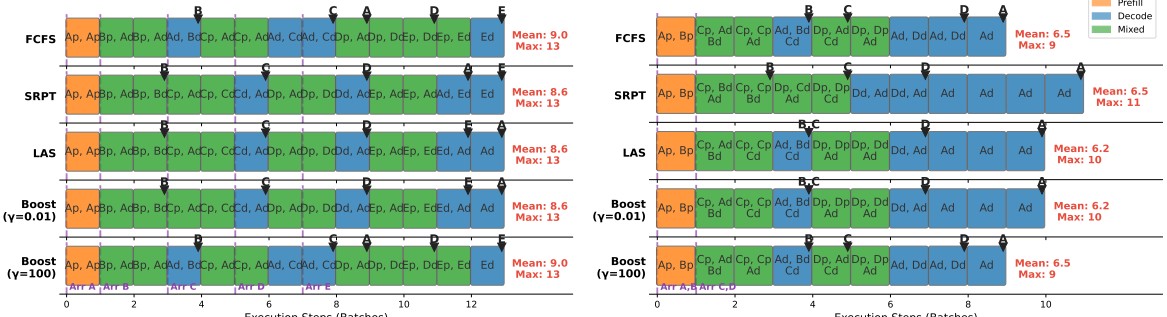

*Figure 3.* **Scheduling Trade-offs and the Boost Scheduler. (Left) The Head-of-Line (HoL) Effect (FCFS weakness):** When short jobs ($B, C, D, E$) arrive behind a long request ($A$), FCFS suffers from blocking, inflating Mean Latency (9.0). SRPT and LAS preempt $A$ to service short jobs first, lowering the mean (8.6). **(Right) Preemption Penalties (SRPT weakness):** In bursty scenarios, strict prioritization by SRPT repeatedly preempts the long request ($A$), delaying its completion significantly. This harms tail latency compared to FCFS without improving Mean Latency. **No Dominant Winner:** FCFS fails on mean latency during convoys, while SRPT/LAS fail on tail latency during bursts. The **Boost Scheduler** smooths this trade-off by interpolating between these extremes using the $\gamma$ parameter; high $\gamma$ (100) mimics FCFS to protect the tail, while low $\gamma$ (0.01) mimics LAS to optimize mean latency.

maintains better fairness. Across the two scenarios, we choose different values of $\gamma$ in the Boost policy to interpolate between these extremes and improve tail latency.

> **Takeaway**
>
> No single LLM scheduling policy dominates: the best policy depends on the *job size distribution* and *arrival burstiness*. Optimizing for **tail latency and throughput** requires scheduler to adaptively interpolate between two extremes, and balance HoL-blocking mitigation (preempt long jobs) against starvation protection (guard long jobs).

**Challenge #2: Need for Cost-aware LLM Schedulers.** Another challenge for SRPT-style scheduling in LLM serving arises from KV-cache-coupled execution and fairness. First, SJF/SRPT may lead to starvation for long-running requests. Unlike previous fairness-promoting design (Sheng et al., 2024), which focuses on fairness between different clients, LTR proposes a *max_waiting_time* fairness metric to evaluate fairness at the per-request level to prevent starvation due to SJF (Fu et al., 2024). One limitation is that *max_waiting_time* is dominated by a single worst inter-token stall: transient batching/memory events can inflate

*max_waiting_time* and trigger quantum promotions even when a request is not persistently starved, which may destabilize batching/KV locality and reduce overall throughput under load.

During decoding, requests accumulate large key–value (KV) caches that must be preserved across preemption, making late-stage interruption costly. Systems such as Sarathi-Serve and PagedAttention focus on improving throughput–latency tradeoffs and memory efficiency, but do not directly address tail-optimal scheduling under preemption overheads (Agrawal et al., 2024a; Kwon et al., 2023). While some prediction-based schedulers introduce heuristics to limit preemption, they do not fundamentally mitigate the brittleness of prediction-driven priority inversion under workload variability.

**This Work: Bridging Queuing with LLM System Optimization.** Unlike prior LLM schedulers based on SJF or SRPT (e.g. LTR and TRAIL (Shahout et al., 2024; Fu et al., 2024)), which primarily optimize mean end-to-end latency or mean TTFT and rely on decoding-size or length rank prediction, our approach explicitly targets tail-latency metrics, including high-percentile TTFT and TTLT. *This distinction is fundamental: while SJF/SRPT are mean-optimal, they offer no guarantees on tail behavior, especially under*

*bursty loads, diverse request input/output length distribution and continuous batching with memory constraints.* Table 1 shows the key distinctions in our work w.r.t workload distribution (heavy-tailed, light-tailed), key assumptions (no size prediction) and the design (preemption-aware, tail-aware, TLLT-aware).

## 3. Formulation

**Notation.** Consider a stochastic service system (LLM engine like SGLang or vLLM) with request arrival process $\{N(t), t \geq 0\}$ governed by rate $\lambda$. Let $\mathcal{R}$ denote the set of requests, where each request $i \in \mathcal{R}$ is characterized by a tuple $(a_i, S_i, \mathbf{s}_i)$:

- $a_i \in \mathbb{R}_+$: arrival time
- $S_i \in \mathbb{R}_+$: intrinsic service requirement (prefill length known at arrival, decode length unknown)
- $\mathbf{s}_i = (s_i^{\text{pre}}, s_i^{\text{dec}}) \in \mathbb{R}_+^2$: decomposed service phases, where $S_i = s_i^{\text{pre}} + s_i^{\text{dec}}$

Let $C_i(\pi)$ denote the completion time of request $i$ under policy $\pi$. The response time is $T_i(\pi) = C_i(\pi) - a_i$.

**Tail Optimization.** LLM serving requires both minimizing the likelihood of extreme latency events (SLO violation rate) and tail latency at large quantiles (P99, P95 latency). We adopt the rigorous definition of *Tail Optimality* relative to the set of feasible policies $\Pi_{\texttt{blind}}$ that operate without exact job size information.

**Definition 3.1** (Tail Optimality Constant). For a scheduling policy $\pi \in \Pi_{\texttt{blind}}$, the optimality constant is defined as:

$$K_\pi = \sup_{\pi^* \in \Pi_{\texttt{blind}}} \limsup_{t \to \infty} \frac{\mathbb{P}[T_\pi > t]}{\mathbb{P}[T_{\pi^*} > t]} \quad (1)$$

Our objective is to design a scheduling policy $\pi_{\text{boost}}$ that achieves a near-optimal tail constant for TTLT and TTFT metrics, ensuring asymptotic equivalence with the optimal blind policy even under stochastic decode lengths.

**The $\gamma$-Boost Priority Function.** To achieve strong tail optimality for response latency distribution, we employ the generalized $\gamma$-Boost score function (Yu and Scully, 2024). For a request with attained service $w_i(t)$ at time $t$, define the *boost function*:

$$b_\gamma(w) = \frac{1}{\gamma} \log \left( \frac{1}{1 - e^{-\gamma w}} \right), \quad \gamma > 0 \quad (2)$$

The priority score assigned to request $i$ at time $t$ is: $\phi_i(t) = a_i - b_\gamma(w_i(t))$

Under appropriate conditions on the service distribution $F_S$ and arrival rate $\lambda$, the policy that schedules the request with minimum $\phi_i(t)$ achieves $K_\pi = 1$ (Harlev et al., 2025), which is strongly-tail-optimal.

- **Forced preemption**: Due to limited GPU memory capacity $M_{\text{GPU}}$, the active KV-cache memory consumption $\sum_{i \in \mathcal{A}(t)} m_i(t)$ is bounded. Preemption overhead is non-constant across decoding phases.
- **Parallel batch processing**: Multiple requests are served simultaneously with continuous batching and scheduling boundaries come at each chunk. The effective service rate will change, violating the M/G/1 assumption.
- **Mix of heavy-tailed and light-tailed distributions**: Coding workload empirically exhibits heavy-tailed decoding length distribution, violating the conditions for tail-optimality in Boost.

We therefore require a scheduler that is simultaneously (i) tail-aware and prediction-free, (ii) robust to workload/burstiness shifts, (iii) cache-/memory-aware under KV-capacity constraints, and (iv) low-overhead to implement in a production continuous-batching pipeline. Our design diagram is shown in Fig. 4.

### 3.1. Phase 1: Prefill-Boosted Strawman DISTBOOST

Inspired by Sarathi's chunked prefill, our strawman approach, DISTBOOST, treats the prefill and decode phases as disjoint scheduling problems.

**Formulation.** Partition the request set into two queues: $Q_p(t) = \{i : w_i(t) < s_i^{\text{pre}}\}$ (prefill) and $Q_d(t) = \{i : w_i(t) \geq s_i^{\text{pre}}\}$ (decode). Define separate priority functions:

$$\phi_i^{\text{pre}}(t) = a_i - b_\gamma(s_i^{\text{pre}}) \quad \forall i \in Q_p(t), \quad (3)$$
$$\phi_i^{\text{dec}}(t) = a_i^{\text{dec}} \quad \forall i \in Q_d(t), \quad (4)$$

where $a_i^{\text{dec}}$ is the time request $i$ enters the decode queue.

The scheduler operates in batched iterations with capacity $B_{\max}$. At each iteration, it selects:

1. **Decode batch**: Select up to $B_{\max}$ requests from $Q_d(t)$ ordered by $\phi_i^{\text{dec}}(t)$:

$$\mathcal{B}_d(t) = \{i_1^*, \ldots, i_k^*\} \text{ where } k = \min(|Q_d(t)|, B_{\max}) \quad (5)$$

2. **Prefill piggyback**: If $k < B_{\max}$, fill remaining slots with chunked prefill requests ordered by $\phi_i^{\text{pre}}(t)$, where each prefill is chunked to size $C$ tokens:

$$\mathcal{B}_p(t) = \arg \min_{\substack{S \subseteq Q_p(t) \\ |S| \leq B_{\max} - k}} \sum_{i \in S} \phi_i^{\text{pre}}(t) \quad (6)$$

Each prefill request $i \in \mathcal{B}_p(t)$ processes at most $C$ tokens per iteration, up to max-num-of-tokens per iteration. The final batch is $\mathcal{B}(t) = \mathcal{B}_d(t) \cup \mathcal{B}_p(t)$.

**Limitations.** The split-phase architecture can exhibit inter-queue head-of-line (HOL) blocking. If the decode queue $Q_d(t)$ contains many long-running decode jobs, a burst of

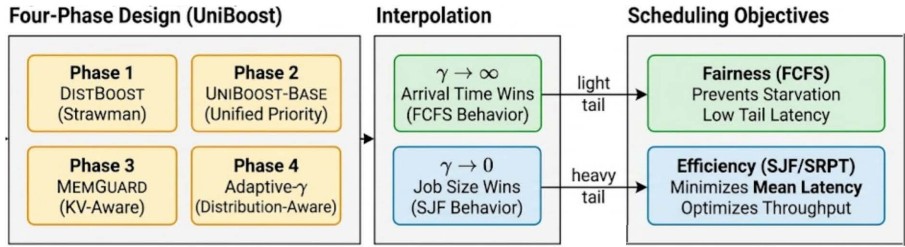

*Figure 4.* The four-phase design, and how it interacts with varying distributions with an asynchronous update to the $\gamma$ parameter.

prefill requests with $s_{i_{j}}^{\text{pre}} \ll \mathbb{E}[s^{\text{dec}}]$ may be delayed because the bandwidth is occupied by decode. Simple heuristics (e.g., promoting certain prefill requests) can reduce this effect, but adaptive, provably safe prioritization is hard to maintain under rapidly changing workloads without risking fairness or throughput degradation.

## 3.2. Phase 2: Unified Prioritization (UNIBOOST-BASE)

To overcome the challenge of cross-phase HoL caused by the Strawman approach, we unify $Q_p$ and $Q_d$ into a single priority space that allows preemption and promotion across phases. Define the *effective work* metric: $\tilde{w}_i(t) = \max(w_i(t), s_i^{\text{pre}})$. The global priority for any request $i$ at time $t$ is: $\Phi(i, t) = a_i - b_\gamma(\tilde{w}_i(t))$

The scheduler selects $i^* = \arg\min_{i \in \mathcal{A}(t)} \Phi(i, t)$, where $\mathcal{A}(t) = Q_p(t) \cup Q_d(t)$ is the active set.

**Preemption Protocol.** When a new high-priority request arrives (or an existing request's priority changes), the system may preempt the currently executing request $i_{\text{curr}}$ if:

$$\exists j \in \mathcal{A}(t) : \Phi(j, t) < \Phi(i_{\text{curr}}, t) - \delta_{\text{hyst}} \qquad (7)$$

where $\delta_{\text{hyst}} > 0$ is a hysteresis parameter (detailed in Section 3.3).

## 3.3. Phase 3: Stability via MEMGUARD (KV-Aware Hysteresis)

Fine-grained priority updates (per-token or per-microbatch) can force frequent context switches when KV-cache capacity is tight. Each switch may evict and reload large KV state, or recomputation. This motivates discretizing when priorities may change to amortize KV movement costs.

**MEMGUARD: quantized priorities + minimum-run hysteresis.** We introduce MEMGUARD, a KV-aware stabilization layer that reduces preemption frequency by discretizing *when* a request's priority may change. Let $w_i$ be the attained decode work (decoded tokens) of request $i$, and let $k$ be a granularity parameter (chosen to align with the KV block size). We define a geometric quantization:

$$\hat{w}_i = k \cdot 2^{\lfloor \log_2(\max\{w_i, k\}/k) \rfloor} \in \{k, 2k, 4k, \ldots\}. \quad (8)$$

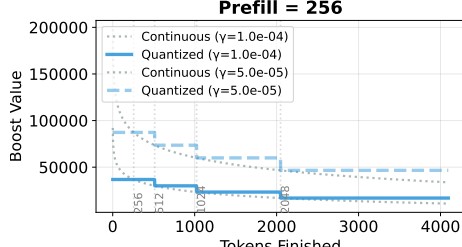

*Figure 5.* An illustration of MemGuard behavior on the boost function. While $\gamma$ is controlled in the outer loop, the more tokens finished, the more "staleness" we assign to that request's hysteresis.

The scheduler recomputes $\Phi(i)$ only when $\hat{w}_i$ changes (i.e., when $w_i$ crosses a threshold). Moreover, once a request is dispatched, it is *non-evictable* until it reaches the next threshold, ensuring a minimum quantum of useful service to amortize KV movement costs.

**Implication: logarithmic preemption opportunities.** Because thresholds grow geometrically, request $i$ can trigger at most $1 + \lfloor \log_2(S_i/k) \rfloor$ priority-revision points before completion, yielding a logarithmic bound on the number of times it can be reconsidered for preemption as a function of its decode length $S_i$. Fig. 5 shows an illustration with $k = 256$. For a 10k decode length, the number is at most 6 swap opportunities under $k = 256$.

## 3.4. Phase 4: Adaptive Parameter Estimation ($\gamma$-Ada)

The optimal boost parameter $\gamma^*$ in Equation 2 depends on the arrival rate $\lambda$, server utilization $\rho$, and the tail index of the job size distribution. A static or arbitrarily chosen $\gamma$ under varying load and distribution leads to sub-optimal tail behavior.

**Online Estimation.** When the observed response time $T$ exhibits a lighter tail, we estimate an *effective* exponential tail rate over the upper tail band $[t_{95}, t_{99}]$ by fitting a log-linear slope: $\ln \bar{F}_T(t) \approx a - \gamma_{\text{eff}} t, \quad t \in [t_{95}, t_{99}]$, yielding

$$\hat{\gamma}_{t+1} = -\frac{\ln \bar{F}_T(t_{99}) - \ln \bar{F}_T(t_{95})}{t_{99} - t_{95}} \qquad (9)$$

This $\gamma_{\text{eff}}$ should be interpreted as a local tail-slope summary statistic. Here $t_p = \inf\{t : \bar{F}_T(t) \le (100 - p)/100\}$ is the empirical $p$-th percentile latency.

Putting these together, we summarize UNIBOOST and its algorithm in Appendix Algo 1.

# 4. Evaluation

**Experimental Setups.** We compare UNIBOOST against state-of-the-art baselines and ablate each design component. UNIBOOST achieves the best overall trade-off, reducing tail TTFT/TTLT by $37\%-60\%$ across various workloads while improving throughput by $1.01-1.12\times$ under bursty load.

**Testbed.** The end-to-end evaluation testbed is a NVIDIA-A100 DGX server with 8 NVIDIA A100 80GB GPUs, 96 vCPUs, and 248GB host memory. The backends are using paged-attention and chunk prefill by default. We disabled the radix cache and prefix caching to do a controlled study on the scheduler.

**Serving Models.** We use a variety of model families including Llama-3-8B, CodeLlama-34B and QWen-72B. For QWen-72B we use TP=4 and CP=4. To stress memory-constrained settings, we use CodeLlama-34B under TP=1. To stress dynamic load, we use Llama-3-8B with dynamic QPS scaled from Azure coding trace 9. All experiments use FP16/BF16 precision.

**Workloads.** We evaluate using the Azure Conversation and ShareGPT (Team, 2023; Microsoft Azure, 2024) for conversation and s1k (Muennighoff et al., 2025) datasets for reasoning. Azure traces also include each request's timestamp, which we scale to preserve the arrival bursty behavior. For workloads without timestamps, we evaluate with Poisson arrival with different QPS. We sample 10k requests for evaluation to reduce warm-up and tear-down impact. We also created two mixed-type workloads, when an LLM engine is receiving both reasoning and non-reasoning requests (mix-1 has 20% reasoning and mix-2 has 70% reasoning, the rest sampled from Azure function, or AZF traces).

**Baselines.** We compare with several state-of-the-art baselines: **1) Sarathi**: Chunk-prefill, decode-prioritized continuous batching, with round-robin ordering in the decode and piggy-back prefills. This policy is integrated in the latest release of vLLM (v1) and SGLang (v0.5.8) for its favorable latency-throughput tradeoffs; **2) SJF (LTR+)**: LTR with perfect prediction. Improved SJF with LTR's starvation prevention techniques using quantum control (Fu et al., 2024). **3) LAS (MLFQ+)**: least-attained service: this is a variant of non-clairvoyant Foreground-Background policy commonly used in operating systems, evolving from discrete MLFQ with finite levels to its idealized continuous form. However, its performance doesn't have a guarantee in light-tailed distribution (Scully et al., 2018). **4) SRPT (TRAIL+)**: TRAIL with perfect prediction. Idealized version of SRPT with TRAIL's swap-prevention threshold after 0.6 decoding length. This policy, like FastServe's Skip-join

MLFQ (MLFQ+), is mean-optimal but not tail under the assumption of heavy-tailed distribution (Shahout et al., 2024).

## 4.1. Comparison Over Baseline Policies

To quantify scheduler performance, Table 2 computes relative improvements compared to TRAIL+ (with perfect knowledge of decoding length) as the oracle baseline. The setup is using Llama-8B with a mixture of prompts of varying lengths, 70% from reasoning tasks and 30% from Azure traces. Percentage changes are computed as latency: $\frac{\text{baseline}-\text{value}}{\text{baseline}} \times 100\%$, throughput: $\frac{\text{value}-\text{baseline}}{\text{baseline}} \times 100\%$. Positive values indicate improvement over TRAIL+.

We observe from the table that while all schedulers exhibit some degradation in latency metrics compared to TRAIL+, UNIBOOST trades off some mean latency in this workload but demonstrates the most substantial gains across all major tail metrics, especially in P99 TTLT and TTFT. DIST-BOOST, the strawman version of our approach, also shows good improvements, particularly in P99 TTLT, but degrades in TTFT, the reason being that it prioritizes decoding queue over prefill. Sarathi shows decreases in all metrics, and SJF, while still improving over SRPT in TTLT tail, experience pronounced degradation because of its lack of preemption to reclaim KV memory for shorter requests, and its natural focus for heavy-tailed workloads. Overall, UNIBOOST stands out as the most effective scheduler in this comparison, delivering significant performance enhancements across the board, improving P99 TTLT around 35% and P95 TTFT around 97%, without sacrificing throughput.

## 4.2. Analysis and Robustness of Improvement Across Workloads

Fig. 6 plots the *percent slowdown* of each baseline compared to UNIBOOST on a reasoning workload with QWen-72B. Three consistent patterns emerge.

**(1) TTFT is where most baselines lose first, and the loss accelerates in the upper tail.** This behavior is consistent with a *prefill admission* pathology: when prefill is not explicitly protected, prefill work gets repeatedly delayed by decode-heavy microbatches, so any burst of long-running (reasoning) decodes increases the waiting time before a request can even start producing the first token. UNIBOOST avoids this by enforcing a distribution-aware priority that prevents prefill from being perpetually deferred by decode occupancy, which is especially important when reasoning requests create long decode residency times.

**(2) TTLT gaps come from different tail failure modes: instability under long decodes vs. convoying.** In the middle panel, all baselines remain above 0% across percentiles, but they diverge sharply near the tail: LAS or SRPT-like

*Table 2.* Relative performance compared to TRAIL (baseline). Green indicates improvement, yellow indicates minor degradation ($\leq 15\%$), orange indicates moderate degradation ($\leq 50\%$), red indicates severe degradation ($> 50\%$).

| | End-to-End Latency | | | TTFT | | | TBT | | | Throughput |
|---|---|---|---|---|---|---|---|---|---|---|
| **Scheduler** | Mean | P95 | P99 | Mean | P95 | P99 | Mean | P95 | P99 | tok/s |
| **TRAIL+** | +0.0% | +0.0% | +0.0% | +0.0% | +0.0% | +0.0% | +0.0% | +0.0% | +0.0% | +0.0% |
| **SJF** | -11.1% | -12.5% | +11.2% | -74.6% | -70.5% | -9.7% | -5.2% | -7.1% | -6.3% | -7.9% |
| **Sarathi** | -12.3% | -13.2% | -7.6% | -49.9% | -40.5% | -8.1% | -6.8% | -8.4% | -7.9% | -10.9% |
| **DistBoost** | -14.0% | +6.0% | +37.1% | -22.8% | -35.0% | -3.4% | -14.1% | -10.2% | +18.3% | +1.1% |
| **UniBoost** | -19.1% | +1.1% | +35.1% | +52.1% | +97.4% | +34.0% | -25.3% | -8.1% | +33.8% | +1.2% |

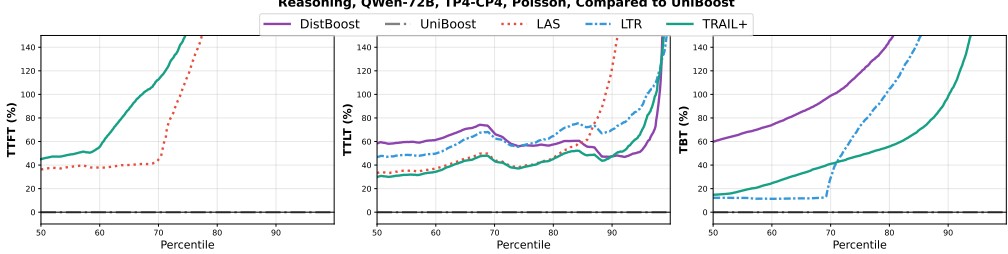

*Figure 6.* (Higher is Better) Percentile latency gap (median P50 to tail P99.9) of End-to-End Latency (TTLT), Time to First Token (TTFT) and TBT compared with the UNIBOOST policy. Every policy uses continuous batching with chunked-prefill (chunk size=1024) under high ($\rho = 0.99$) system-attainable load. UNIBOOST demonstrates significant latency reduction at the tail.

policy spikes dramatically in the high percentiles. This is a classic *attained-service inversion* under memory-coupled decoding: LAS prioritizes jobs with smaller attained service, so newly-arrived (or recently-unblocked) requests repeatedly jump ahead of partially-served long decoding requests. It creates more *simultaneously active* requests, increasing paging/eviction pressure and causing additional stalls. The result is a positive feedback loop: more interleaving → more KV pressure → longer residency → less throughput and more queuing → worse tail TTLT. UNIBOOST avoids this by smoothly interpolating between serving earlier arrivals and shorter jobs, reducing reshuffles and tail oscillations.

**(3) TBT shows decode-prioritization alone is insufficient: mixing increases variance.** In the right panel, DISTBOOST is uniformly slow, while LTR stays flat then degrades after ~P70. Two mechanisms explain this phenomenon: (i) over-prioritizing decodes raises the number of concurrent decoders, increasing per-token contention that harms TBT broadly; (ii) thresholded/length-ranked switching (LTR) is brittle—once a load/mixture threshold is crossed it flips regimes, boosting mixing and heterogeneity and triggering tail blow-ups.

**Takeaway.** UNIBOOST remains near the Pareto frontier because it controls the prefill–decode balance in each iteration and the degree of interleaving with a single smooth control knob, which stabilizes both TTFT admission and tail TTLT/TBT under long reasoning decodes.

### 4.3. Ablation of Each Design Phase

Fig. 7 sweeps QPS with Codellama-34B on 1 GPU. The load–latency curve exhibits a sharp knee: Among all the

variants, DISTBOOST (purple) destabilizes first (0.24 QPS), with P90/P99 rising from hundreds of ms to seconds. Changing it to Phase-2 (blue) delays the knee and roughly halves P99 near 0.248 QPS. Adding MEMGUARD in Phase-3 (green) further suppresses KV-swap thrashing, cutting tails by another 1.5-2×. UNIBOOST (orange) performs best: at 0.255 QPS P99 remains sub-second while alternatives are multi-second. Overall, UNIBOOST shifts the knee right by 4–6% and reduces P90/P99/mean by up to 10× near saturation compared with baseline MLFQ+. These plots jointly illustrate the effectiveness on MEMGUARD and ADA mechanism, especially in protecting high-load latency. The exact timeline of $\gamma$ change is shown in Appendix Fig. 9.

### 4.4. SLO Attainment Across Scales and Datasets

For LLM service provider, the goal is to find the maximum SLO scale the system can tolerate while still achieving the attainment target (99% or 95%). For TTFT, CombinedBoost (UNIBOOST interchangeably) maintains highest attainment across the widest range of SLO scales. In contrast, SJF exhibits the lowest tolerance to stringent SLOs, with TTFT attainment degrading rapidly as the SLO tightens. On average, 2.9–8.7 × more stringent SLO can be achieved with UNIBOOST, with mixture-2 slightly showing larger gap due to its more dynamic nature. For TTLT, it is more interesting to note that although there is a general 1.7–4.3× gain in SLO threshold that can be achieved with 99% attainment, DistBoost does a little better, by trading off the TTFT attainment to prioritize decoding.

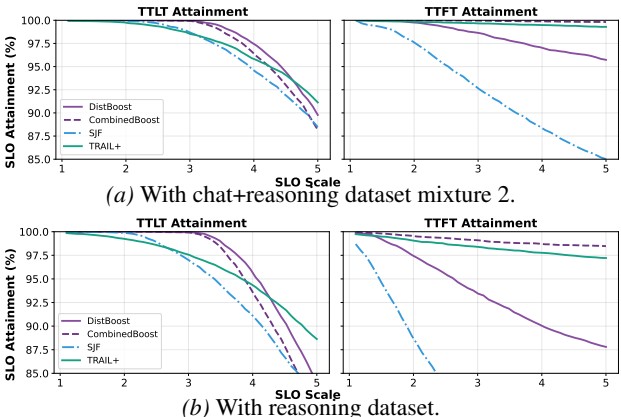

*Figure 7.* Design-phase ablation using load-latency plots.

*(a)* With chat+reasoning dataset mixture 2.

*(b)* With reasoning dataset.

*Figure 8.* (Larger is better) SLO attainment with different SLO scales. Note: Base SLO scale=1 is chosen with Sarathi's P95 latency under 0.9 max load. X-axis is in reciprocal scale (SLO = Base/$x$).

## 5. Conclusion and Future Work

We presented UNIBOOST, a prediction-free, tail-aware scheduling framework for LLM serving that uses soft, continuous priority shaping and co-designs scheduling with KV-cost–aware preemption to improve tail latency under diverse workloads. Inspired by theory, it shows superior performance on tail under heavy load, and even beats prediction-based policy with only a lightweight runtime statistic tracker. The implementation is simple to integrate into existing serving stack with continuous batching, p/d disaggregation or chunk-prefilled systems like vLLM and SGLang. The results are shown in Appendix §A.5.

Future work includes extending UNIBOOST to multi-model and multi-replica deployments (joint routing + caching + scheduling), incorporating richer memory signals, developing formal guarantees that capture preemption/eviction costs in modern LLM inference systems, and leveraging recent design insights on optimizing tail latency in heavy-tailed queues (Li et al., 2026) to better handle reasoning workloads with highly variable output lengths.

## Acknowledgments

Yueying Li and Edward Suh were supported by the NSF under grant CCF-2118709. Yueying Li acknowledges Anvil AI and GPU allocations through allocation CIS230253 from the Advanced Cyberinfrastructure Coordination Ecosystem: Services & Support (ACCESS) program, which is supported by U.S. National Science Foundation grants #2138259, #2138286, #2138307, #2137603, and #2138296. Udit Gupta was supported by NSF Grant CCF-232660 and CFF-2326608, and acknowledges support from Google and Amazon. Ziv Scully was supported by the NSF under grant nos. CMMI-2307008 and CCF-2544452.

## Impact Statement

This paper presents work aimed at improving the efficiency of LLM inference. There are many potential societal consequences of increasing LLM efficiency, none of which we feel must be specifically highlighted here.

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

# A. Appendix

## A.1. Algorithm details

**The full implementation of UNIBOOST and its helper functions (sketch) is shown below.**

## A.2. Theoretical guarantee of no starvation

We adopt notation and terminology similar to that of Yu and Scully (2024). Consider any moment in time, which we will call time 0. The *crossing work* for time 0, denoted $V$, is, roughly, the amount of work that "boosts past time 0". Specifically, $V$ is the sum of, for each request A that arrives after time 0, the amount of processing time A receives during which it has boosted arrival time before time 0.

The reason $V$ is relevant to starvation is that (a variant of) the argument in Section 3 of Yu and Scully (2024) implies, roughly speaking, that the worst-case scenario for any given request's latency under Boost is to have the same latency as under FCFS, plus at most $V$. (See their Lemma 3.3 and eqs. (3.2–3.3), though these have additional terms that turn out to be negligible but obscure the aforementioned takeaway.) This means $V$ bounds the worst-case degradation of Boost compared to FCFS, even for the largest jobs. Specifically, in order for $V$ to not significantly degrade tail performance, we need $\mathbf{E}[e^{\gamma V}]$ to be finite. Lemma 3.5 of Yu and Scully (2024) shows that for size-based boost functions, this holds if the boost function $b(s)$ is (weakly) decreasing and satisfies $b(s) \leq O(s^{-1})$ as $s \to 0$. Below, we show that nearly the same result holds for attained-service-based boost functions.

**Lemma A.1.** *Consider a Boost policy in an M/G/1 queue that gives a job of attained service $a$ boost at most $b(a)$. Suppose $b(a)$ is (weakly) decreasing and that there exists*

$\epsilon > 0$ *such that $b(a) \leq O(a^{-(1-\epsilon)})$ as $a \to 0$. Then $\mathbf{E}[e^{\gamma V}] < \infty$.*

Before giving the proof sketch, we note that the boost function precondition applies to UniBoost, because it assigns a job of attained service $a$ a boost of at most

$$b_\gamma\left(\tfrac{1}{2}a\right) = \frac{1}{\gamma} \log \frac{1}{1 - e^{-\frac{1}{2}\gamma a}} \leq O(\log a^{-1}) \leq O(a^{-0.01}).$$

The assumption of an M/G/1 queue is admittedly limiting, but M/G/1 analysis of Boost policies is the state-of-the-art in queueing theory, so this type of result is the best one can hope for at the moment. We expect the argument to extend to the M/G/k, which is closer to a good model of LLM service with continuous batching. But much less is known about Boost in the M/G/k (Yu et al., 2025), so the argument would go beyond a simple extension of a known result, and it would not account for the idiosyncrasies of the parallelizable prefill phase.

*Proof sketch.* We assume without loss of generality that $b(a)$ has an inverse $b^{-1}(t)$ (as one can perturb the "flat parts" of $b(a)$ to make it invertible). We start by following most of the proof of Lemma 3.5 of Yu and Scully (2024), but with one key change:

- In their setting of size-based boosts, a job of size $s$ that arrives at time $t$ contributes $s\,\mathbf{1}(s < b^{-1}(t))$ work towards $V$.

- But in our setting of attained-service-based boosts, a job of size $s$ and attained service $a$ that arrives at time $t$ contributes $\min(s, b^{-1}(t))$ work towards $V$.

Applying Campbell's and Tonelli's theorems as in their proof thus yields, writing $S$ for the size of a random arrival and $\lambda$ for the arrival rate,

$$\mathbf{E}[e^{\gamma V}] = \exp\left(\lambda\,\mathbf{E}\left[\int_0^\infty \left(e^{\gamma \min(S, b^{-1}(t))} - 1\right) \mathrm{d}t\right]\right).$$

From here, it is just a matter of simplifying the integral, which uses the precondition $b(a) \leq O(1/a^{1-\epsilon})$ and the known fact that $\mathbf{E}[e^{\gamma S}] < \infty$. This part is quite different from Yu and Scully's proof, so we give the full computation:

$$\mathbf{E}[e^{\gamma V}] = \exp\left(\lambda\,\mathbf{E}\left[\int_0^\infty \left(e^{\gamma \min(S, b^{-1}(t))} - 1\right) \mathrm{d}t\right]\right)$$

$$= \exp\left(\lambda\,\mathbf{E}\left[\int_0^\infty \int_0^{\min(S, b^{-1}(t))} \gamma e^{\gamma u}\, \mathrm{d}u\, \mathrm{d}t\right]\right)$$

$$= \exp\left(\lambda\,\mathbf{E}\left[\int_0^\infty \int_0^S \gamma e^{\gamma u}\, \mathbf{1}(u < b^{-1}(t))\, \mathrm{d}u\, \mathrm{d}t\right]\right)$$

$$= \exp\left(\lambda\,\mathbf{E}\left[\int_0^S \int_0^\infty \gamma e^{\gamma u}\, \mathbf{1}(t < b(u))\, \mathrm{d}t\, \mathrm{d}u\right]\right)$$

---

**Algorithm 1** UNIBOOST: Adaptive Tail-Aware Scheduling with MEMGUARD

---

**Require:** KV capacity $B$, micro-batch cap $M$, chunk size $c$, bin size $k$, initial $\gamma$
1: Maintain two heaps: $\mathcal{Q}_{\text{pre}}$ (prefill-ready), $\mathcal{Q}_{\text{dec}}$ (decode-ready), keyed by **priority** $\pi(\cdot)$
2: Maintain lightweight runtime stats $\mathcal{S}$ (e.g., recent tail quantiles)
3: **while** server is running **do**
4:    $\gamma \leftarrow$ UPDATEGAMMA($\mathcal{S}$)           $\triangleright$ adaptive tail controller
5:    **for all** active request $r$ (waiting or running) **do**
6:       $s_r \leftarrow$ SIGNAL($r, \mathcal{S}$)        $\triangleright$ prediction-free size signal (phase-aware)
7:       $\tilde{s}_r \leftarrow$ QUANTIZE($s_r, k$)       $\triangleright$ MEMGUARD: geometric bins
8:       $b_r \leftarrow$ BOOST($\tilde{s}_r, \gamma$)
9:       $\pi(r) \leftarrow a_r - b_r$       $\triangleright$ $a_r =$ arrival time (or phase-entry time); smaller $\pi$ served earlier
10:      UPDATEKEY($r, \pi(r)$) in the appropriate heap ($\mathcal{Q}_{\text{pre}}$ or $\mathcal{Q}_{\text{dec}}$)
11:    **end for**
12:    $\mathcal{B} \leftarrow \emptyset, U \leftarrow$ current KV usage
13:    **while** $|\mathcal{B}| < M$ **do**
14:       $r \leftarrow \arg\min\{\text{TOP}(\mathcal{Q}_{\text{pre}}), \text{TOP}(\mathcal{Q}_{\text{dec}})\}$       $\triangleright$ unified ordering across phases
15:       **if** $r$ is None **then**
16:          **break**
17:       **end if**
18:       $\Delta U \leftarrow$ KVNEED($r, c$)       $\triangleright$ KV needed for next prefill chunk or decode step
19:       **if** $U + \Delta U \leq B$ **then**
20:          POP($r$) from its heap; add $r$ to batch $\mathcal{B}$; $U \leftarrow U + \Delta U$
21:       **else**
22:          $v \leftarrow$ SELECTVICTIM(running set, $\pi$, swap-cost)
23:          SWAPOUT($v$); $U \leftarrow U -$ KVRESIDENT($v$)   $\triangleright$ MEMGUARD ensures each $r$ triggers $\leq 1 + \lfloor\log_2(S_r/k)\rfloor$ priority updates/swaps
24:       **end if**
25:    **end while**
26:    EXECUTEONESTEP($\mathcal{B}, c$)       $\triangleright$ run prefill chunks + decode step; update KV + token counters
27:    UPDATESTATS($\mathcal{S}$, observed TTFT/TTLT/TBT)
28: **end while**

---

$$= \exp\left(\lambda \mathbf{E}\left[\int_0^S \gamma e^{\gamma u}\, b(u)\, du\right]\right)$$

$$\leq \exp\left(\lambda \mathbf{E}\left[\int_0^1 \gamma e^\gamma\, b(u)\, du + \int_0^S \gamma e^{\gamma u}\, b(1)\, du\right]\right)$$

$$\leq \exp\left(\lambda\gamma e^\gamma \int_0^1 O(u^{-(1-\epsilon)})\, du + \lambda\, b(1)\, \mathbf{E}[e^{\gamma S} - 1]\right)$$

$$< \infty. \qquad\qquad \square$$

*Table 3.* Short-request E2E latency (seconds).

|  | Mean | P50 | P90 | P95 | P99 |
|---|---|---|---|---|---|
| UniBoost | 2.80 | 2.85 | 3.28 | 3.41 | 3.62 |
| Sarathi | 2.83 | 2.87 | 3.35 | 3.50 | 3.84 |
| SJF | 2.80 | 2.86 | 3.27 | 3.39 | 3.62 |
| SRPT | 2.80 | 2.86 | 3.27 | 3.39 | 3.69 |

### A.3. Starvation Experiment

**Workload.** 90% short requests ($\sim$125 prefill, 100 decode) + 10% long requests (2k prefill, 6k decode). Poisson arrivals at $\rho \approx 0.99$. Llama-3-8B on 8 H100 with random load balancing policy.

We also counted the preemption time of longer requests to capture starvation statistics:

**Key Takeaways.** For each scheduler, we fit a linear regression of long-request E2E latency against request arrival order (Request ID). A scheduler exhibits starvation if later-arriving long requests experience systematically worse latency than earlier ones—i.e., the queue "remembers" and penalizes long requests over time.

1. **Aligned with theory intuition**: slopes are negligible ($\sim$0.0003 s/req), confirming long-request latency does not grow with arrival order (Table 4). Under $\rho < 1$, no scheduler exhibits starvation—consistent with the three-term decomposition bound where crossing work $W_j^{\text{cross}}$ has exponentially decaying tails.

2. **UniBoost matches baselines in fairness, excels in tail**

```
 1: function BOOST(x, γ)                                                              ▷ γ-boost (any soft boosting curve works)
 2:     return 1/γ log( 1/(1−e^{−γx}) )
 3: end function
 4: function QUANTIZE(s, k)                                                           ▷ geometric bins anchored at k
 5:     return k · 2^{⌊log₂(max{1,s/k})⌋}
 6: end function
 7: function SIGNAL(r, S)                                                             ▷ prediction-free, phase-aware
 8:     if r in prefill then
 9:         return prompt tokens (known)
10:     else
11:         return attained decode tokens
12:     end if
13: end function
14: function UPDATEGAMMA(S)
Require: recent E2E latency samples L (window), smoothing β, bounds [γ_min, γ_max], ε
15:     x̂₉₅ ← QUANTILE(L, 0.95);   x̂₉₉ ← QUANTILE(L, 0.99)
16:     Δ ← max(x̂₉₉ − x̂₉₅, ε)                                                        ▷ avoid instability if tail compresses
17:     γ̂ ← (log(99) − log(95))/Δ
18:     γ ← (1 − β) · γ + β · γ̂                                                       ▷ EMA smoothing
19:     return clip(γ, γ_min, γ_max)
20: end function
```

Table 4. Long-request E2E latency (seconds).

|          | Mean | P90   | P95   | P99   | Max   |
|----------|------|-------|-------|-------|-------|
| UniBoost | 88.5 | 185.5 | 190.2 | 200.5 | 245.9 |
| Sarathi  | 88.5 | 184.8 | 190.0 | 199.7 | 205.5 |
| SJF      | 88.4 | 182.8 | 188.0 | 206.9 | 253.9 |
| SRPT     | 88.3 | 183.9 | 188.6 | 196.5 | 246.3 |

Table 5. Long-request preemption P99 (seconds).

|          | P99  |
|----------|------|
| UniBoost | 9.1  |
| Sarathi  | 6.6  |
| SJF      | 13.3 |
| SRPT     | 18.6 |

**latency for shorter requests**: UniBoost achieves comparable long-request E2E latency (Table 4) and preemption overhead (Table 5) to SRPT and SJF—schedulers that require knowing job sizes *a priori*—without any size oracle, while delivering competitive short-request tail latency (Table 3). This makes it practical for autoregressive LLM serving where decode length is unknown at arrival.

## A.4. Hyperparameter Sensitivity

UniBoost has three main hyperparameters; their roles are orthogonal:

- **Adaptive** $\gamma$ controls the fairness–responsiveness trade-off (FCFS-like vs. short-job-first-like). We note that it is already self-tuned in our design (Section 3.4 and Figure 9), and the tuning frequency is adjusted with respect to the workload and arrival rate (by collecting sufficient samples). We observe stable performance across a broad range of starting values, provided we allow adaptive tuning in response to changes in workload distribution.

- **Bin size** $k$ (Section 3.3) is the granularity that controls stability vs. agility in reprioritization; larger $k$ causes less thrashing, especially under higher load, but may lead to longer residence and less agility for smaller requests to chime in ($k=1$ is just the vanilla boost). We chose a value of 256 due to the GPU's internal tile structure (Table 6). This is orthogonal to chunk size of the scheduler as it only controls the memory backend stickiness.

- **Hysteresis** $\delta$ (Section 3.2) prevents pathological micro-preemption. Empirically, we find that this works well under a broad range of values (Table 7), likely due to the $k$ we already set.

We further provide sensitivity studies on these parameters in Tables 6 and 7.

## A.5. Interaction with other system optimization

**Prefix/Radix Caching**   Caching is outside the scope of this work, but UniBoost can be extended straightforwardly

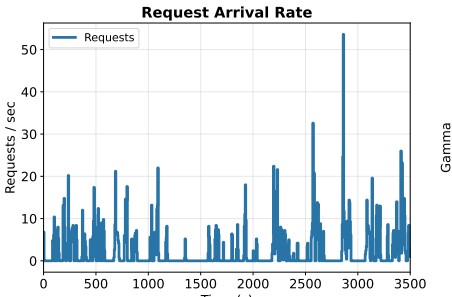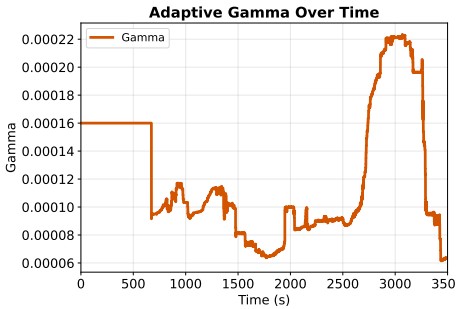

*Figure 9.* Bursty arrival in the Azure Function Traces and the corresponding $\gamma$ changes across time (scaled).

*Table 6.* Sensitivity to bin size $k$.

| $k$ | P99 E2E (ms) | P99 TTFT (ms) | P99 TBT (ms) | Mean TBT (ms) | Throughput (req/s) |
|---|---|---|---|---|---|
| 64 tokens | 8530 | 392 | 112 | 55 | 3.94 |
| 256 tokens (default) | 8512 | 390 | 115 | 54 | 3.95 |
| 512 tokens | 8681 | 403 | 128 | 54 | 3.93 |
| 1024 tokens | 8465 | 450 | 137 | 53 | 3.90 |

to include prefix caching. Caching primarily reduces *effective prefill work*. If a request has $h_i$ cached tokens, we simply replace the formula in effective work as

$$\tilde{w}_i(t) = \max\big(w_i(t),\ s_i^{\text{pre}} - h_i\big),$$

where $w_i(t)$ is the total number of tokens processed so far. Thus the UniBoost implementation and its validity remain unchanged; we only need to plug in the expected hit tokens (which is usually the sum of tokens from earlier turns for a multi-turn workload).

**Chunked Prefill and Continuous Batching**  Our design already adopts continuous batching and chunked prefill (Lines 341–345): prefill is treated as incremental service that contributes to the same unified signal $\tilde{w}_i(t)$. Each chunk advances the effective work and therefore updates priority consistently with decode. The chunk size is typically set based on the desired TBT/TPOT SLO, so we use 1024 for CodeLlama. We provide additional chunk values in Table 8 to demonstrate that our algorithm performs well across any SLO regime.

Service with continuous batching is similar to the classical M/G/$k$ model from queueing theory: there are $k$ slots, and the jobs in each slot can be swapped in and out independently of each other. There is theory about the Boost policy for the M/G/$k$ (Yu et al., 2025), but it is limited, especially for jobs with unknown or partially known sizes. What little theory exists suggests that the Boost policy framework is effective in the M/G/$k$, though the details of how to optimally assign boost amounts are are not fully known.

**Composability with Global Dispatch Policies**  Cluster-level routers (e.g., load balancers, global schedulers) decide which instance a request is sent to; UniBoost decides how to schedule requests *within* that instance. When the routing policy is exogenous with respect to the GPUs' states—which may not hold exactly for a load-aware policy but is a reasonable approximation when the number of servers is large—the per-instance scheduling guarantees of UniBoost carry over directly. Moreover, combining UniBoost with a stronger global policy can yield further SLO improvements in an iso-resource setting.

Table 9 reports the relative improvement of UniBoost over the vLLM baseline (vLLM v1.0—Sarathi with chunked-prefill continuous batching) under three routing strategies on 64 H100 GPUs.

With JSQ (join-shortest-queue) routing—which provides better load balancing per instance—the TBT and E2E improvements become more pronounced, confirming that UniBoost composes favorably with stronger global dispatch policies.

**Scalability with Cluster Size**  To evaluate scalability, we use trace-replay methodology—standard in the systems community for emulating large clusters. Traces are down-sampled with timestamps at the global router, and the resulting instance-level requests are replayed to emulate clusters of varying size with a random dispatcher. Table 10 reports the relative improvement of UniBoost over baseline vLLM (lower is better) on a mixed reasoning + chat workload. The gains remain consistent across scales from $32\times$ to $512\times$, indicating that UniBoost's per-instance benefits are largely independent of cluster size.

*Table 7.* Sensitivity to hysteresis $\delta$.

| $\delta$ | P99 E2E (ms) | P99 TTFT (ms) | P99 TBT (ms) | Mean TBT (ms) | Throughput (req/s) |
|---|---|---|---|---|---|
| 0 (none) | 9500 | 385 | 105 | 72 | 3.80 |
| 0.1 (default) | 8512 | 390 | 115 | 54 | 3.95 |
| 0.3 | 8561 | 392 | 122 | 55 | 3.94 |
| 0.5 (aggressive) | 8575 | 394 | 121 | 54 | 3.94 |

*Table 8.* Sensitivity to chunk size. Values are relative change (%) vs. the FCFS baseline.

| Chunk size | E2E | | | TTFT | | | TBT | | | tok/s |
|---|---|---|---|---|---|---|---|---|---|---|
| | Mean | P95 | P99 | Mean | P95 | P99 | Mean | P95 | P99 | |
| 128 | −17.8 | +2.4 | +38.6 | +12.3 | +18.5 | +8.2 | −22.1 | −5.9 | +36.4 | +0.6 |
| 512 | −18.7 | +1.5 | +36.0 | +35.8 | +58.1 | +22.3 | −24.5 | −7.5 | +34.8 | +1.0 |
| **1024** | **−19.1** | **+1.1** | **+35.1** | **+52.1** | **+97.4** | **+34.0** | **−25.3** | **−8.1** | **+33.8** | **+1.2** |
| 2048 | −19.4 | +0.8 | +34.5 | +68.4 | +93.6 | +45.1 | −26.0 | −8.6 | +33.2 | +1.3 |

*Table 9.* Relative improvement of UniBoost over vLLM under different global schedulers ($64\times$ H100).

| Global Scheduler | Scheduler | Avg E2E | P90 E2E | P99 E2E | Avg TBT | P90 TBT | P99 TBT |
|---|---|---|---|---|---|---|---|
| Random | UniBoost | +2.1% | −3.7% | −13.9% | −68.8% | −17.0% | −69.5% |
| Round-robin | UniBoost | +2.7% | −0.9% | −13.5% | −48.0% | +5.5% | −60.5% |
| JSQ | UniBoost | −23.0% | −17.2% | −23.5% | −73.5% | −52.6% | −77.5% |

*Table 10.* Relative improvement of UniBoost over vLLM at different cluster scales.

| Scale | Avg E2E | P99 E2E | Avg TTFT | P99 TTFT |
|---|---|---|---|---|
| $32\times$ | −2.3% | −13.9% | +8.4% | −8.7% |
| $64\times$ | −2.8% | −24.2% | +3.6% | −10.3% |
| $128\times$ | −1.9% | −12.7% | +7.8% | −11.9% |
| $512\times$ | −0.4% | −23.2% | +1.2% | −13.5% |

