# Beyond Prediction: Tail-Aware Scheduling for LLM Inference

## Abstract

LLM serving exhibits extreme length variability, making size-based scheduling difficult in practice. Recent LLM schedulers approximate SJF/SRPT using predicted decode lengths or rank and primarily report mean-centric metrics (e.g., TTFT/TBT). We show these prediction-driven policies can be fragile under distribution shifts, bursty arrivals, and GPU memory pressure, and still offer limited control over tail latency (P90–P99) that dominates user experience—even with perfect decode-length knowledge. We introduce a distribution-aware, prediction-free scheduling framework that replaces explicit length prediction with soft, $\gamma$-parameterized priority boosting driven by lightweight statistical signals. Our design co-optimizes scheduling with cache-aware preemption to account for memory-coupled decode dynamics that vary across workload mixes. Evaluated on Azure production traces, our method achieves a P99 TTLT up to 35-50% lower than SRPT with perfect length prediction and a TTFT 34-47% lower across various workloads, including reasoning-heavy and chat-heavy tasks, demonstrating a robust alternative for tail-latency optimization in online LLM serving.

## 1. Introduction

Large language models (LLMs) are increasingly deployed in interactive applications such as code generation, conversation, and agentic workloads. In these settings, user experience is governed not only by average performance but also by tail latencies (often defined as 99% or 95% percentile TTFT, TTLT and TBT (Agrawal et al., 2025; Zhong et al., 2024)), making resource allocation and scheduling an important area to balance throughput and user experience (Duan et al., 2024; Sun et al., 2024; Luo et al., 2025; Zhang et al., 2025b).

Beyond *global* resource allocation like sharding, disaggregation, and routing (Chen et al., 2025; Shi et al., 2025; Patke et al., 2024), modern LLM *local* replica schedulers play an important role in balancing throughput and latency trade-offs by considering the right batch-size, priority, and

eviction order (Agrawal et al., 2024a;b). Recently, more advanced scheduling systems in LLM often adopt *prediction-based scheduling*, motivated by classical results showing that Shortest Job First (SJF) or Shortest Remaining Processing Time (SRPT) minimize *mean* response time when job sizes are known (Qiu et al., 2024; Fu et al., 2024; Shahout et al., 2024; Srivatsa et al., 2024). They predict output length or the number of remaining tokens and approximate SJF/SRPT in practice. While effective at reducing average latency under accurate prediction, these methods are largely optimized for mean-centric metrics, and can exhibit poor tail latency behaviors (Scully et al., 2018; Nair et al., 2010; Nuyens et al., 2008).

This challenge is amplified by the rise of *reasoning-augmented LLMs*, where generation may involve multi-step reasoning, self-reflection, tool invocation, or adaptive termination. As a result, decode lengths are highly variable. Two requests with identical prefill sizes may diverge by orders of magnitude in completion time, and the job-size distribution can shift rapidly across workloads and over time. This inherent unpredictability makes it difficult to design schedulers that rely on job size estimation or request ranking.

These observations raise a key question: *do schedulers need job lengths or rankings predictions to achieve tail-optimal latency?* Rather than attempting to predict decoding length or ranking, we argue that scheduling should be guided by running statistics and optimized for the tail adaptively.

Our approach is inspired by recent advances in *tail-optimal scheduling theory*, which studies how to reduce asymptotic tail waiting times (e.g., P95/P99). A key takeaway is that, instead of *hard* size-based ranking (e.g., SJF/SRPT), one can use *soft, continuous* priority shaping: each request receives a smoothly varying priority score computed from lightweight signals.

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

 (Scully et al., 2020). More recently, Harlev et al. develop a Gittins-index policy that explicitly targets tail latency optimization under stochastic service dynamics

(Harlev et al., 2025). Complementary to these results, Yu and Scully establish strong tail optimality for the $\gamma$-Boost policy in light-tailed M/G/1 queues, showing that soft, continuous boosting improves not only the exponential decay rate but also the tail constant of response time distributions (Yu and Scully, 2024). These works highlight a fundamental distinction between mean optimality and tail optimality, motivating scheduling designs that go beyond average latency.