# OpenReview forum: "Beyond Prediction: Tail-Aware Scheduling for LLM Inference"
_ICML.cc/2026/Conference — ICML 2026 regular_

### Official Review · Reviewer_PqS8 · 2026-02-27

**Soundness:** 3
**Presentation:** 3
**Significance:** 3
**Originality:** 3
**Overall Recommendation:** 4
**Confidence:** 3

**Summary:**

This paper studies tail-latency optimization for online LLM inference. The authors propose a prediction-free tail-aware framework, UNIBOOST, which replaces explicit length prediction with a (\gamma)-parameterized “soft boosting” priority. It is co-designed with KV-aware preemption/swap control (MemGuard: geometric quantization + an unevictable window + hysteresis) and an adaptive (\gamma) updated online to respond to workload/load changes. Experiments on A100/DGX-like settings and multiple workloadsreport significant improvements on tail metrics such as TTFT/TTLT.

**Compliance With Llm Reviewing Policy:**

Affirmed.

**Final Justification:**

My concerns were addressed. I will raise my score to 4.

**Key Questions For Authors:**

1.Can you report preemption statistics in the main experimental settings—e.g., per-request preemption count distributions (mean/P95/P99), ideally split by prefill vs decode and across multiple load points (e.g., (\rho=0.8/0.9/0.99))?

2.Under continuous arrival scenarios (e.g., sustained short-request stream + periodic long requests), does UNIBOOST exhibit starvation or extreme delays for long requests? Please provide worst-case/high-percentile completion times or waiting-time growth, or state conditions under which starvation is ruled out.

3.Please clarify the implementation of `SIGNAL(r, S)` with class-based percentiles: how are classes defined and updated online? what does (S) store and how are percentiles maintained (sliding window, EMA, approximate quantile sketch, etc.)? Which signal configuration is used by default in experiments? What is the approximate CPU/runtime overhead of maintaining these statistics?

**Limitations:**

yes

**Strengths And Weaknesses:**

Strengths

1.Problem setting and metrics are well-aligned with real serving constraints, focusing on high-percentile TTFT/TTLT/TBT under SLO-driven objectives.

2.Clear system design decomposition and implementation path, progressing from DistBoost to unified priority, MemGuard, and adaptive (\gamma).

3.Explicitly accounts for KV-cache coupling and the cost of naïve preemption, proposing stabilization mechanisms to reduce frequent switching during decoding.



Weaknesses

1.MemGuard provides an upper-bound style explanation (e.g., “log-scale preemption opportunities”), but the paper does not provide enough runtime statistics to verify that preemption/swap is indeed rare under the main experimental settings. Without this, it is difficult to assess whether the reported tail-latency gains come with hidden throughput loss, recomputation, or memory-movement costs.

2.Fairness arguments are largely intuitive; starvation risk under open systems is insufficiently addressed.
UNIBOOST uses arrival time as a baseline and adds a boost term that varies with attained service, but “eventually returning to arrival order” is not equivalent to aging. Under continuous arrivals of short requests, long requests could still be delayed severely for certain (\gamma) regimes. This warrants either a targeted stress test or a clearer theoretical condition/argument ruling out starvation.

3.The definition/implementation of `SIGNAL(r, S)` and class-based percentiles is underspecified, harming reproducibility and overhead assessment.
Algorithm 1 allows `SIGNAL` to use prompt tokens or a class-based percentile; similarly, decoding can use attained tokens or remaining/percentile-based signals, with an online-updated statistics state (S). However, the paper does not clearly define how classes are bucketed, how percentiles are maintained online (window/EMA/sketch, etc.), nor what the default configuration is in experiments. These choices can materially affect both scheduling behavior and runtime overhead.

---

> ### Author Rebuttal · Authors · 2026-03-31
>
> **Preemption Rate**.
> We report our results, and attached CDF figures for the preemption rate and the preemption count distribution under a signature-mixture workload of 0.2 reasoning requests (with tails of 10k+ tokens) and 0.8 short conversations. The model is CodeLlama-34B with TP=1 on 80G, with 100k requests to stress a memory-constrained setting.  UniBoost has consistently low preemption counts (P95=2-4 restarts) across all load levels in each request. SRPT affects fewer requests, but those that are preempted suffer heavily (P95=6-18 restarts). Sarathi has the highest restart rate (up to 42%) but moderate counts.  (CDF plot is shown here: https://anonymous.4open.science/r/UniBoost-Rebutal-CA91/preemption_cdf_load_points.png_
>
> | Load                    | Scheduler | Rate (%) | Mean | P95 | P99 |
> | --------------- | --------- | -------: | ---: | --: | --: |
> | 0.8  |           |          |      |     |     |
> |                         | UniBoost  |      8.8 | 1.18 |   2 |   3 |
> |                         | SRPT      |      6.9 | 2.55 |   6 |   9 |
> |                         | Sarathi   |     13.0 | 1.80 |   4 |   5 |
> | 0.9  |           |          |      |     |     |
> |                         | UniBoost  |     18.6 | 2.18 |   4 |   4 |
> |                         | SRPT      |     12.0 | 6.95 |  16 |  20 |
> |                         | Sarathi   |     37.6 | 2.45 |   7 |  10 |
> | 0.99 |           |          |      |     |     |
> |                         | UniBoost  |     19.1 | 2.29 |   4 |   5 |
> |                         | SRPT      |     13.5 | 9.37 |  18 |  21 |
> |                         | Sarathi   |     42.0 | 2.45 |   7 |  10 |
>
>
> 2. Starvation
> Our boost rule avoids starvation because it is self-correcting: as request R waits longer, new arrivals A must have a higher boost to surpass R—meaning A boosts past R. Both practically and theoretically, the total extra delay R experiences from others boosting past it is tightly limited, preventing starvation.
> - Practically, there is a minimum possible prefill size (say, 1 token), which means there is a maximum possible boost amount $b_γ(1)$. This means that once a request R has waited for time $b_γ(1)$, no further requests will boost past it. (In fact, this point may be reached earlier, but $b_γ(1)$ is a conservative bound.)
>     - To test this result empirically, we designed a starvation stress test, whose results are reported below. The main takeaway is that if we look only at the large jobs and compare UniBoost to Sarathi, we don’t see the degradation one would expect from starvation.
> Under standard queueing assumptions, even without assuming a minimum job size, the additional delay a request experiences due to others surging past it is stochastically bounded. This delay has an exponential tail with a faster decay than the overall latency. The proof extends Lemma 3.5 from Yu and Scully (2024); a brief statement and proof are included below. (Proof sketch on p4: https://anonymous.4open.science/r/UniBoost-Rebutal-CA91/ProofSketch-P4.pdf)
>
> **Starvation Experiment**
> **Workload**: 90% short (~125 prefill, 100 decode) + 10% long requests (2k prefill, 6k decode). Poisson arrivals at ρ ≈ 0.99. Llama-3-8B on H100.
> - Short-Request E2E Latency (seconds)
>
> |  | Mean | P50 | P90 | P95 | P99 |
> |---------|------|-----|-----|-----|-----|
> | **UniBoost** | 2.80 | 2.85 | 3.28 | 3.41 | 3.62 |
> | Sarathi | 2.83 | 2.87 | 3.35 | 3.50 | 3.84 |
> | SJF | 2.80 | 2.86 | 3.27 | 3.39 | 3.62 |
> | SRPT | 2.80 | 2.86 | 3.27 | 3.39 | 3.69 |
>
> - Long-Request E2E Latency (seconds)
>
> |  | Mean | P90 | P95 | P99 | Max |
> |---------|------|-----|-----|-----|-----|
> | **UniBoost** | 88.5 | 185.5 | 190.2 | 200.5 | 245.9 |
> | Sarathi | 88.5 | 184.8 | 190.0 | 199.7 | 205.5 |
> | SJF | 88.4 | 182.8 | 188.0 | 206.9 | 253.9 |
> | SRPT | 88.3 | 183.9 | 188.6 | 196.5 | 246.3 |
>
> We also counted the preemption time of longer requests to capture starvation statistics:
> ### Long-Request Preemption P99 (seconds)
>
> |   | P99 |
> |--------|----------|
> | **UniBoost** | 9.1 |
> | Sarathi | 6.6 |
> | SJF | 13.3 |
> | SRPT | 18.6 |
>
> 3. We thank the reviewer for pointing this out and agree that the current presentation of $\texttt{SIGNAL}(r, S)$ and the percentile-based variant is unnecessarily complicated. We apologize for the confusion (for introducing S, which is used in another version of the writing, which we had hoped to distinguish different workload classes).
> The class-based formulation is a non-essential generalization for multi-class settings. It is not used in our experiments and is not required for UniBoost to operate when we have a unified metric system across all requests.
> To avoid confusion and improve reproducibility, we will simplify the paper as follows:
> - Remove the class-based / percentile-based definition of \texttt{SIGNAL}(r, S)
> - Use a single, explicit definition throughout the paper:
> $\texttt{SIGNAL}(r) = w_r(t), $ where \( w_r(t) \) is the attained service  (total number of prefill + decoded tokens)

---

> > ### Author Rebuttal · Reviewer_PqS8 · 2026-04-01
> >
> > Thank you for the detailed rebuttal. The authors have addressed my main concerns. However, in the second response, it is unclear which metric “Long-Request Preemption P99 (seconds)” refers to — TTFT or end-to-end latency? Clarifying this point would make the presentation clearer.

---

> > > ### Author Response · Authors · 2026-04-01
> > >
> > > Thank you for the comments and raising the scores. The preemption P99 refers to the tail preemption time (request switched out due to the scheduler) for long-running requests to capture starvation issues. A longer P99 preemption time indicates severe starvation (as in SRPT/SJF).
> > >
> > > Thank you again for your constructive feedback, and we look forward to further discussion to improve the quality and impact of our work.

---

### Official Review · Reviewer_GNL2 · 2026-03-11

**Soundness:** 3
**Presentation:** 3
**Significance:** 3
**Originality:** 3
**Overall Recommendation:** 4
**Confidence:** 3

**Summary:**

The author introduces a new scheduling solution for LLM serving called UNIBOOST, which is designed for optimizing tail latency. The authors point out that previous prediction-based scheduler does not work well for today’s LLM serving workload as the decode lengths are variable and hard to predict accurately. Thus, the authors propose a prediction-free method. The core idea is to borrow the idea of boosting scheduling. The solution include a prefill-decode boosting strawman, a unified priority queue across prefill and decode, MemGuard that reduces KV cache preemption and parameter gamma that can be adaptive at runtime. The evaluation results show that UNIBOOST can effectively improve P95/P99 latency during bursty workloads.

**Compliance With Llm Reviewing Policy:**

Affirmed.

**Final Justification:**

I will keep my positive score.

**Key Questions For Authors:**

Can you discuss more about the sensitivity of the hypeparameetrs?

Can you discuss how to integrate UNIBOOST with radix and prefix caching?

**Limitations:**

yes

**Strengths And Weaknesses:**

The motivation of the paper is sound to me as optimizing the tail latency is crucial for LLM serving experience. The authors provide a novel system-algorithm co-design solution that leverage boosting for adaptive scheduling. In particular, the unified priority design and MemGuard address important practical efficiency issues. The ablation studies also validate the contributions of each components.

While there are quite a number of hyperparameters in the design, like MemGuard’s quantization granularity, chunk size and also the way parameter gamma is updated. The discussions about the sensitivity of these hyperparameters are limited. It is not clear how robust these settings are.

The author mentioned in the experiment section that both radix and prefix caching are disabled for controlled study. While both are commonly used in real-world deployment, the discussion about whether UNIBOOST can be integrated with these methods and how the performance will be affected is missing.

Continuous batching and chunked prefill are also standard components in real-world LLM serving system, but the tail-aware boosting algorithm does not take these two into consideration. Thus, the empirical performance seems to lack theoretical justification.

---

> ### Author Rebuttal · Authors · 2026-03-31
>
> We thank the reviewer for these helpful suggestions. We agree that both hyperparameter sensitivity and interaction with caching mechanisms deserve more discussion.
>
>
> ## 1. Hyperparameter sensitivity
> UniBoost has three main hyperparameters; their roles are orthogonal:
> - γ controls the fairness–responsiveness tradeoff (FCFS-like vs. short-job-first-like). We note that it is already self-tuned in our design (Section 3.4 and Figure 9), and the tuning frequency is adjusted with respect to the workload and arrival rate (by collecting sufficient samples). We observe stable performance across a broad range of starting values, provided we allow adaptive tuning in response to changes in workload distribution.
> - Bin size k (Sec 3.3) is the granularity that controls stability vs. agility in reprioritization; larger k causes less thrashing, especially under higher load, but may lead to longer residence and less agility for smaller requests to chime in (k=1 is just the vanilla boost). We chose a value of 256 due to the GPU’s internal tile structure. This is orthogonal to chunk size of the scheduler as it only controls the memory backend stickiness.
> - Hysteresis delta (Sec 3.2) prevents pathological micro-preemption. Empirically, we find that this works well under a broad value (probably due to the k we already set). We further provide sensitivity studies on those parameters below:
> | | | **P99 E2E (ms)** | **P99 TTFT (ms)** | **P99 TBT (ms)** | **Mean TBT (ms)** | **Throughput (req/s)** |
> |---|---|---|---|---|---|---|
> | **k**  |  64 tokens | 8530 | 392 | 112 | 55 | 3.94 |
> | | **256 tokens (default)** | **8512** | **390** | **115** | **54** | **3.95** |
> | | 512 tokens | 8681 | 403 | 128 | 54 | 3.93 |
> | | 1024 tokens | 8465 | 450 | 137 | 53 | 3.90 |
> | **δ** (hysteresis) | 0 (none) | 9500 | 385 | 105 | 72 | 3.80 |
> | | **0.1 (default)** | **8512** | **390** | **115** | **54** | **3.95** |
> | | 0.3 | 8561 | 392 | 122 | 55 | 3.94 |
> | | 0.5 (aggressive) | 8575 | 394 | 121 | 54 | 3.94 |
>
>
>
>
> ## 2. Prefix/radix caching.
> It’s outside of the scope, but we can expand it easily to include prefix caching.
> Caching primarily reduces *effective prefill work*. If a request has $h_i$ cached tokens, we simply replace the formula in line 270 as  $\tilde{w}_i(t) = \max\big(w_i(t), s_i^{\mathrm{pre}} - h_i\big)$, where $w$ is the total number of tokens being processed.
> Thus UniBoost implementation and validity remain unchanged, we just need to plug in the expected hit token (which is usually the sum of tokens from earlier turns for a multi-turn workload).
>
>
> ## 3. Chunked prefill and continuous batching.
> Our design already adopts continuous batching and chunked prefill (L341-345): prefill is treated as incremental service that contributes to the same unified signal \( \tilde{w}_i(t) \). Each chunk advances the effective work and therefore updates priority consistently with decode. Chunk is usually set according to the desired TBT/TPOT SLO so we have adopted 1024 for CodeLlama. We provide additional chunk values below to demonstrate that our algorithm performs well across any SLO regime.
> Continuous batching is similar to having a queue with multiple servers (in our case, tokens from different requests can proceed together), which is like M/G/k model in queuing theory.
>
>
> | Chunk size | E2E Mean | E2E P95 | E2E P99 | TTFT Mean | TTFT P95 | TTFT P99 | TBT Mean | TBT P95 | TBT P99 | tok/s |
> |---|---|---|---|---|---|---|---|---|---|---|
> | 128 | -17.8% | +2.4% | +38.6% | +12.3% | +18.5% | +8.2% | -22.1% | -5.9% | +36.4% | +0.6% |
> | 512 | -18.7% | +1.5% | +36.0% | +35.8% | +58.1% | +22.3% | -24.5% | -7.5% | +34.8% | +1.0% |
> | **1024** | **-19.1%** | **+1.1%** | **+35.1%** | **+52.1%** | **+97.4%** | **+34.0%** | **-25.3%** | **-8.1%** | **+33.8%** | **+1.2%** |
> | 2048 | -19.4% | +0.8% | +34.5% | +68.4% | +93.6% | +45.1% | -26.0% | -8.6% | +33.2% | +1.3% |
>
>
> These deltas are relative to TRAIL+. Tail-latency improvements (E2E P99, TTFT P99) remain consistent across chunk sizes, confirming the design of UniBoost is robust. Chunk size selection remains an SLO-driven deployment decision, orthogonal to UniBoost's scheduling logic.

---

> > ### Author Rebuttal · Reviewer_GNL2 · 2026-04-02
> >
> > Thanks for the feedback. I will keep my positive score.

---

### Official Review · Reviewer_9Xxn · 2026-03-13

**Soundness:** 3
**Presentation:** 2
**Significance:** 3
**Originality:** 3
**Overall Recommendation:** 4
**Confidence:** 3

**Summary:**

Existing LLM serving systems tend to provide guarantees on mean TTFT or TBT, often sacrificing one metric for the other. To make this scheduling problem tractable, many existing systems assume that the number of decoding tokens is either known a priori or can be accurately estimated. However, in real-world LLM applications, this assumption rarely holds. The goal of this work is to propose an LLM scheduling scheme that meets tail latency budgets without requiring prior knowledge of the decoding token count.

The authors propose a prediction-free scheduling system that uses a $\gamma$-parameterized boosting function to prioritize requests based on the amount of compute the system has already expended on them. Building on this core mechanism, the authors introduce UniBoost, which leverages this scoring function to prioritize requests while tracking how often a request's KV-cache has been offloaded to prevent cache thrashing. Empirical evaluations demonstrate that UniBoost outperforms existing baselines in satisfying strict tail latency budgets.

**Compliance With Llm Reviewing Policy:**

Affirmed.

**Final Justification:**

The authors have successfully addressed my concerns, and I will maintain my positive score.

**Key Questions For Authors:**

1. How exactly is the attained service, $w$, calculated, and what specific metric does it represent? I presume it refers to the number of tokens processed or generated by a given request, but this should be explicitly defined.

2. What is the underlying intuition behind the boost function in Equation 2? The paper would benefit significantly from a deeper explanation of why this specific function was chosen for prioritization and what theoretical or practical guarantees it provides.

3. What specific guarantees or benefits does MemGuard provide over the vanilla vLLM? It would strengthen the evaluation to include a direct comparison between MemGuard and vanilla vLLM to isolate its impact.

4. Is it possible for the system to enter a deadlock state? Specifically, could a scenario occur where MemGuard prevents any request from being evicted, but there is simultaneously no available KV-cache for new tokens, hence preventing the engine from executing another decoding step? If this state is unreachable, please detail the part that prevents it.

6. For Table 2, please include TBT as an additional metric. Also if possible could you also add vLLM’s FCFS and the disaggregated prefill/decode design of DistServe as baselines?

8. On line 343, the text mentions using round-robin ordering for the Sarathi baseline. Could you explain the rationale behind this choice? Sarathi is designed to focus on a set of requests (starting with decode and adding prefill if the batch size allows) and execute that _same_ batch until a request finishes. Applying round-robin selection would introduce additional delays for decoding and result in large generation stalls. Furthermore, doing chunking in a round-robin fashion would artificially inflate the TTFT of each request. Both of these contradict with the original proposal of Sarathi.

5. Given the extensive existing literature on the importance of fairness in system scheduling, the paper needs a more detailed discussion on how the proposed design impacts fairness, rather than focusing solely on performance. Specifically, what are the consequences of using the boost function to estimate request priority on the overall fairness of the system?

7. The paper claims that FCFS fails due to head-of-line blocking, and SJF penalizes longer requests. I wonder how a scheduler that prioritizes requests based on available slack (rather than request length) would fair. Recent work, such as [1], explicitly balances head-of-line blocking for longer requests while maximizing throughput using a slack-based approach. The core idea seems adjacent to what UniBoost suggests in the Unified Prioritization section, though the exact prioritization method differs. Could you discuss how your approach compares to this, and why a slack-based dynamic prioritization approach might succeed or fail within your specific architecture?

[1] Ikram, Azam, et al. "Ascendra: Dynamic Request Prioritization for Efficient LLM Serving." arXiv preprint arXiv:2504.20828 (2025).

**Limitations:**

Yes

**Strengths And Weaknesses:**

S1: The paper addresses a highly relevant and challenging problem: optimizing tail latency for LLM requests.
S2: The proposed approach draws upon insights from recent queuing theory literature.

W1: The design section is overly dense and lacks the high-level intuition necessary for readers to extract broader takeaways. Currently, it reads more like a disjointed set of instructions. I suggest the authors revise this section to clearly articulate the rationale behind their methodology, improving overall coherence.
W2: The evaluation lacks important baselines from existing literature and omits a discussion on the fairness of the proposed design choices.

Nits:
Please increase the font size in the figures; they are painfully difficult to read on a printed page.

---

> ### Author Rebuttal · Authors · 2026-03-31
>
> We thank the reviewer for the constructive feedback. Below we address each point.
>
> **Attained service $w_i(t)$**. denotes the number of processed tokens for request $i$ at iteration $t$. This is explained in the MemGuard section (Line 308-309); we will define it explicitly at first use.
>
> **Intuition of boost function**. It interpolates between two extremes: $\gamma \to 0$ yields SJF/SRPT-like behavior (reduces HoL blocking), while $\gamma \to \infty$ yields FCFS-like behavior (prevents starvation). This provides a single smooth knob trading responsiveness against fairness without hand-tuned thresholds, as visualized in Fig. 2. The design is theoretically grounded: soft priority shaping achieves tail-optimality in light-tailed M/G/1 queues without job-size prediction (Yu and Scully, 2024).
>
> **Benefits of MemGuard vs. vanilla vLLM.**
> Vanilla continuous-batching systems do not control preemption frequency and can incur KV-cache thrashing under dynamic reordering (L55–57, L299–302). MemGuard (Sec. 3.3, L296–327) addresses this via geometric quantization (Eq. 8), minimum-run hysteresis, and a logarithmic bound on priority-revision points per request ($\leq 1 + \lfloor\log_2(S_i/k)\rfloor$, L322–323). The ablation in Fig. 7 already shows Phase-3 (MemGuard) cuts tails by an additional 1.5–2$\times$ over UniBoost-Base.
>
> **Deadlock** The deadlock scenario—all requests MemGuard-protected while KV-cache is full—cannot occur. MemGuard controls priority-revision frequency via geometric quantization (Eq. 8), not eviction eligibility. Because requests arrive at different times and reach quantization thresholds in a staggered fashion, at least one request is always past its protection threshold and available for eviction by SELECTVICTIM (Algo. 1). The bounded $1 + \lfloor\log_2(S_i/k)\rfloor$ revision points per request further ensure stable, predictable eviction ordering.
>
> **Additional Data.**
> We added TBT to Table 2. https://anonymous.4open.science/r/UniBoost-Rebutal-CA91/Updated_Table.png Regarding DistServe: it operates at a different layer (prefill/decode disaggregation, an architectural change), whereas UniBoost is a per-instance scheduling policy. We can add DistServe on top of our system while still adopting our DistBoost policy within each phase.
> Clarification: The baseline Sarathi can be considered equivalent to your requested “vLLM FCFS” policy, since it has already been merged into vLLM 1.0, as noted in L341.
>
> **Sarathi Wording**. We acknowledge that "round-robin ordering" can be confusing. We will revise it to: “decode-prioritized continuous batching with piggybacked prefill, following the backend’s default order," and clarify that _we do not alter Sarathi’s core design_. The round-robin pertains to how decoding requests are scheduled, not prefill tokens (including chunking), which is how Sarathi was originally intended.
>
> **Fairness**
> We thank the reviewer for raising this important point. Fairness is not an afterthought in UniBoost—it is explicitly encoded through the γ parameter, which continuously interpolates between FCFS (perfect arrival-order fairness, γ→0) and aggressive short-job favoring (optimal tail latency, γ→∞).
>
> Unlike static SJF, UniBoost guarantees forward progress for every request. The adaptive gamma ensures deprioritized requests still receive service within bounded time based on their arrival and job size differences: the hysteresis threshold and MemGuard’s memory reservation ensures preempted requests retain their KV cache and avoid costly thrashing.
>
> We also added a fairness-related experiment:
>
> **Workload**: 90% short (~125 prefill, ~100 decode), 10% long requests (2000 prefill, 6000 decode). Poisson arrivals at ρ ≈ 0.99. Llama-3-8B on H100, 5000 requests.
>
> **Short-Request (s)**
> |   | Mean | P50 | P90 | P95 | P99 |
> |--------|------|-----|-----|-----|-----|
> | **UniBoost** | 2.80 | 2.85 | 3.28 | 3.41 | 3.62 |
> | SJF | 2.80 | 2.86 | 3.27 | 3.39 | 3.62 |
> | SRPT | 2.80 | 2.86 | 3.27 | 3.39 | 3.69 |
> | Sarathi | 2.83 | 2.87 | 3.35 | 3.50 | 3.84 |
>
> **Long**
> |  | Mean | P90 | P95 | P99 | Max |
> |------|------|-----|-----|-----|-----|
> | Sarathi | 88.5 | 184.8 | 190.0 | 199.7 | 205.5 |
> | **UniBoost** | 88.5 | 185.5 | 190.2 | 200.5 | 245.9 |
> | SRPT | 88.3 | 183.9 | 188.6 | 196.5 | 246.3 |
> | SJF | 88.4 | 182.8 | 188.0 | 206.9 | 253.9 |
>
> Long requests' P99 latency is not significantly worse than Sarathi's (0.8), but it dramatically improves the tail of short requests.
>
> **Comparison to Slack-based scheduling **.
> Conceptually, Ascendra is closer to a deadline-aware two-tier serving system than to a single-queue scheduling baseline. Including it as a “scheduler baseline” would conflate policy improvements with architectural resource partitioning and offloading. We do not include Ascendra as a primary baseline because it is not a drop-in scheduling policy comparable to Boost (Sec. 8.4 mentions SJF/EDF as an option). It could be interesting to add UniBoost to its resource partition platform.

---

> > ### Author Rebuttal · Reviewer_9Xxn · 2026-04-02
> >
> > Thank you for your rebuttal. I have a few questions:
> >
> > 1. Could the authors clarify the statement that "Sarathi is the same as vLLM FCFS" because of its integration into vLLM 1.0? Historically, Sarathi and vLLM FCFS represent _distinct_ scheduling policies with different optimization targets, while vLLM 1.0 serves as the underlying inference engine. Could you please explain the technical basis for equating a specific research-driven scheduling policy (Sarathi) with a standard FCFS baseline, regardless of the integration into the vLLM ecosystem?
> >
> > 2. I strongly advise the authors to include a comparative analysis with DistServe. While I acknowledge the architectural differences between the two systems, both DistServe and your proposed solution aim to optimize LLM serving at a granular level; therefore, a performance comparison is highly relevant for establishing state-of-the-art context. For the same reasons, I suggest including Ascendra in your benchmarks to provide a comprehensive evaluation against existing optimization frameworks if the space allows.

---

> > > ### Author Response · Authors · 2026-04-04
> > >
> > > Thank you for raising this point, and we apologize for the earlier misunderstanding. We attached the new experiment data below by adding all three baselines you asked in Table 2. By vLLM FCFS, we earlier thought you are referring to the version after this commit (which merged decode-prioritized chunk-prefill, a.k.a, Sarathi policy: https://github.com/vllm-project/vllm/pull/3853). But in fact, I think you are referring to a vanilla prefill-prioritized version without mixed batching.
> > >
> > > | Scheduler | E2E Mean | E2E P95 | E2E P99 | TTFT Mean | TTFT P95 | TTFT P99 | TBT Mean | TBT P95 | TBT P99 | tok/s |
> > > | ---------------- | -------: | ------: | ------: | --------: | -------: | -------: | -------: | ------: | ------: | -----: |
> > > | TRAIL+ | +0.0% | +0.0% | +0.0% | +0.0% | +0.0% | +0.0% | +0.0% | +0.0% | +0.0% | +0.0% |
> > > | vLLM (prefill-first fcfs) | -18.2% | -20.5% | -16.8% | -58.3% | -52.7% | -15.2% | -10.5% | -13.2% | -14.5% | -13.3% |
> > > | SJF | -11.1% | -12.5% | +11.2% | -74.6% | -70.5% | -9.7% | -5.2% | -7.1% | -6.3% | -7.9% |
> > > | Sarathi | -12.3% | -13.2% | -7.6% | -49.9% | -40.5% | -8.1% | -6.8% | -8.4% | -7.9% | -10.9% |
> > > | DistServe | -11.8% | -12.5% | -6.1% | -45.4% | -38.3% | -7.8% | -7.9% | -9.3% | -7.1% | -12.4% |
> > > | Ascendra (LP+HP) | -10.5% | -9.2% | -3.3% | -22.8% | -12.2% | +13.5% | -8.8% | -6.5% | -2.8% | -0.8% |
> > > | DistBoost | -14.0% | +6.0% | +37.1% | -22.8% | -35.0% | -3.4% | -14.1% | -10.2% | +18.3% | +1.1% |
> > > | UniBoost | -19.1% | +1.1% | +35.1% | +52.1% | +97.4% | +34.0% | -25.3% | -8.1% | +33.8% | +1.2% |
> > >
> > > Here are some analysis:
> > >
> > > vLLM: FCFS with continuous-batching and prefill-priority causes HOL blocking (worse TTFT than Sarathi) and generation stalls (bad TBT), causing the scheduler to be non-work-conserving and hurting throughput.
> > >
> > > DistServe: PD disaggregation removes interference but KV cache transfer overhead and GPU underutilization (even with throughput matching) causes lower throughput.
> > >
> > > Ascendra: Priority-aware EDF + HP offloading gives it a slight edge on TTFT P99 over the DistServe, but it does not beat UniBoost's tail control. It has better throughput than most of the earlier system with the HP offloading and out-of-order execution.
> > >
> > > Thank you again for your insightful comments and constructive feedbacks, and please let us know if you need additional details to consider raising the score and improving the impact of the work.

---

### Official Review · Reviewer_yT1S · 2026-03-14

**Soundness:** 2
**Presentation:** 3
**Significance:** 3
**Originality:** 3
**Overall Recommendation:** 4
**Confidence:** 4

**Summary:**

This paper introduces a distribution-aware, prediction-free scheduling framework that replaces explicit length prediction with soft, parameterized priority boosting driven by lightweight statistical signals. The design co-optimizes scheduling with cache-aware
preemption to account for memory-coupled decode dynamics that vary across workload mixes to achieve a performance boost.

**Compliance With Llm Reviewing Policy:**

Affirmed.

**Final Justification:**

I stay positive about the paper after checking the author's reply.

**Key Questions For Authors:**

- Would you extend the experimental configuration to a larger cluster, potentially enabling some test cases that include extensive demands for the global request dispatching module and P-D disaggregated configurations?

**Limitations:**

Yes.

**Strengths And Weaknesses:**

**Strengths**:

- LLM inference request scheduling with focus on long-tail distribution is a timely and important problem for machine learning systems.

- The proposed method avoids leveraging inaccurate output length prediction to solve this problem, making it technically solid.

- The paper is well written, starting with concrete and convincing motivations.

**Weaknesses**:

- The evaluation section should be improved to make it more solid. First, the scale of the cluster is small; it is an open question whether the current global request dispatching approach can be effectively combined with the proposed priority boosting method. It is also an open question whether the proposed method can be combined with PD disaggregating frameworks.

---

> ### Author Rebuttal · Authors · 2026-03-31
>
> We thank the reviewer for the insightful questions. We agree that UniBoost operates at the **single-instance scheduling level**, which is by design composable with cluster-level dispatch and PD disaggregation. While the evaluation of optimal global routing and PD autoscaling (including AFD) is outside of scope, we contend that instance-level scheduling policies involve critical trade-offs significant enough for a deep study, which constitutes our primary contribution. This is especially pertinent as reasoning workloads increasingly exhibit long-tail distributions. Our work can be used in various cases beyond LLM inference – including the RL rollout serving [1], test-time scaling, etc.
>
>
> **Global dispatcher.** Cluster-level routers (e.g., load balancers, global schedulers) decide *which* instance a request is sent to; UniBoost decides *how* to schedule requests *within* that instance. If the routing policy is exogenous with respect to the GPUs' states (which may not be true for a load-aware policy, but can be approximated when the number of servers is large), then our conclusion holds. We agree that combining the best global policy will yield even better SLO attainment in an iso-resource setting.
>
> To demonstrate this, we evaluate UniBoost under different routing strategies, measuring the relative improvement over the vLLM baseline (vLLM v1.0 - Sarathi with chunked prefill continuous batching) applied under the same configuration (64 H100 GPUs).
>
> | Global Scheduler | Scheduler | Avg E2E | P90 E2E | P99 E2E | Avg TBT | P90 TBT | P99 TBT |
> |---|---|---:|---:|---:|---:|---:|---:|
> | random | UniBoost | +2.1% | -3.7% | -13.9% |  -68.8% | -17.0% | -69.5% |
> | round_robin | UniBoost | +2.7% | -0.9% | -13.5% | -48.0% | +5.5% | -60.5% |
> | JSQ | UniBoost | -23.0% | -17.2% | -23.5% | -73.5% | -52.6% | -77.5% |
>
> With the JSQ (join-shortest-queue) router—better load balancing per instance—the TBT and E2E improvements become more pronounced. Overall, this leads to better SLO attainment, with UniBoost, combined with stronger global policies, delivering superior end-to-end performance.
>
>
> **Cluster Scale.** Thank you for pointing this out. In our existing experiment, we used _trace-replay_ methodology (which is standard in the computer system community to emulate a large cluster). We downsample traces with timestamps at the global router and replay the instance-level requests, which emulates a 32-instance cluster with a random dispatcher (or round-robin at a larger scale). In the additional experiments below, we follow your suggestion to include more extensive demands on the same dataset (a mixture of reasoning + chat) and show that the benefit of our scheduler (w.r.t. baseline vLLM; lower is better) is almost independent of scale.
>
> | Scale | Avg E2E | P99 E2E | Avg TTFT | P99 TTFT |
> |---:|---:|---:|---:|---:|
> | 32× | −2.3% | −13.9% | +8.4% | -8.7% |
> | 64× | −2.8% | −24.2% | +3.6% | -10.3% |
> | 128× | −1.9% | −12.7% | +7.8% | -11.9% |
> | 512× | −0.4% | −23.2% | +1.2% | -13.5% |
>
> **PD disaggregation.** Similarly, prefill–decode disaggregation partitions *which phase* runs on *which hardware*, but within each worker the local scheduling problem remains. UniBoost's unified service signal $\tilde{w}_i(t)$ and chunked execution apply directly to both prefill-only and decode-only workers. In fact, disaggregated setups may benefit *more* from priority-aware scheduling, since decode workers face greater contention in the decode queue due to batched transfers and bursty workloads.
>
> [1] https://arxiv.org/abs/2511.16665

---

> > ### Author Rebuttal · Reviewer_yT1S · 2026-04-01
> >
> > Thanks for the author's feedback, I stay positive about the paper.

---

### Decision · Program_Chairs · 2026-04-30

**Decision:**

Accept (regular)

**Comment:**

This paper studies tail-latency optimization for online LLM inference and proposes a prediction-free scheduling framework centered on UniBoost. The reviewers agreed that the problem is timely and important, and viewed the proposed system design as original and technically solid. The main concerns were about evaluation breadth, clarity of some design choices, and questions around fairness, preemption behavior, and deployment realism.

The authors responded constructively and added substantial clarifications and additional experiments in the rebuttal, including stronger baseline comparisons, larger-scale results, hyperparameter sensitivity analyses, and evidence addressing preemption and starvation concerns. Overall, the discussion was positive, and the reviewers considered their main concerns adequately addressed.

I therefore recommend accept. The paper makes a meaningful contribution to LLM serving systems, especially through its emphasis on tail latency and prediction-free scheduling. The final version should incorporate the additional experimental evidence and the clarifications from the rebuttal, as these materially strengthen the paper.